# Drosophila embryos allocate lipid droplets to specific lineages to ensure punctual development and redox homeostasis

**Marcus D. Kilwein, T. Kim Dao, Michael A. Welte***

Department of Biology, University of Rochester, Rochester, New York, United States of America

* michael.welte@rochester.edu

## Abstract

Lipid droplets (LDs) are ubiquitous organelles that facilitate neutral lipid storage in cells, including energy-dense triglycerides. They are found in all investigated metazoan embryos where they are thought to provide energy for development. Intriguingly, early embryos of diverse metazoan species asymmetrically allocate LDs amongst cellular lineages, a process which can involve massive intracellular redistribution of LDs. However, the biological reason for asymmetric lineage allocation is unknown. To address this issue, we utilize the Drosophila embryo where the cytoskeletal mechanisms that drive allocation are well characterized. We disrupt allocation by two different means: Loss of the LD protein Jabba results in LDs adhering inappropriately to glycogen granules; loss of Klar alters the activities of the microtubule motors that move LDs. Both mutants cause the same dramatic change in LD tissue inheritance, shifting allocation of the majority of LDs to the yolk cell instead of the incipient epithelium. Embryos with such mislocalized LDs do not fully consume their LDs and are delayed in hatching. Through use of a *dPLIN2* mutant, which appropriately localizes a smaller pool of LDs, we find that failed LD transport and a smaller LD pool affect embryogenesis in a similar manner. Embryos of all three mutants display overlapping changes in their transcriptome and proteome, suggesting that lipid deprivation results in a shared embryonic response and a widespread change in metabolism. Excitingly, we find abundant changes related to redox homeostasis, with many proteins related to glutathione metabolism upregulated. LD deprived embryos have an increase in peroxidized lipids and rely on increased utilization of glutathione-related proteins for survival. Thus, embryos are apparently able to mount a beneficial response upon lipid stress, rewiring their metabolism to survive. In summary, we demonstrate that early embryos allocate LDs into specific lineages for subsequent optimal utilization, thus protecting against oxidative stress and ensuring punctual development.

## Author summary

Embryos of diverse animal species sort their lipid droplets into specific cell lineages early in development. Here we prevent Drosophila embryo's ability to undergo this asymmetric

**Data Availability Statement:** All relevant data are within the manuscript and its Supporting Information files.

**Funding:** This work was supported by National Institutes of Health (www.nih.gov) grants F31 HD100127 (to M. D. K.) and R01 GM102155 (to M. A. W). The funders had no role in study design, data collection and analysis, decision to publish, or preparation of the manuscript.

**Competing interests:** The authors have declared that no competing interests exist.

inheritance through two separate genetic perturbations. We then investigate the effects on subsequent embryogenesis, finding developmental delays and an inability to consume the mislocalized lipid droplets. We find a similar delay for embryos which receive fewer lipid droplets from their mothers. To understand how embryos respond to limited access to lipid droplets, we investigate the global transcriptome and proteome in the mutants and find alterations to the expression of hundreds of metabolism genes, including for enzymes that transport sugar across cell membranes, break down fat, or protect against oxidative stress. We test if the upregulated genes have beneficial functions for lipid-deprived embryos. We find that zygotic knock down of the upregulated oxidative stress response genes GSS and GSTT4 as well as the lipase ATGL lowers hatching success in LD-deprived genetic backgrounds, but not wild type. In summary, we find that early lipid droplet lineage sorting sets the stage for metabolic success in subsequent embryogenesis.

## Introduction

Lipid droplets (LDs) are ubiquitous organelles that store cells' neutral lipid reserves [1,2]. They store crucial lipids, including triglycerides, sterol esters, and other fat-soluble molecules, and are integral to energy homeostasis, lipid signaling processes, and membrane synthesis. They are also abundant in the embryos of many metazoans, including Drosophila, mud snails, Xenopus, zebrafish, and mice, where they presumably fuel embryogenesis [3,4,5,6]. Intriguingly, in many species, embryos allocate their LD reserves asymmetrically across lineages. Whether this asymmetric inheritance could benefit embryonic development or be an epiphenomenon without consequence for the embryo is unknown. Previous studies have relied on aggregate biochemical assays that could not address the role of spatial specialization in embryonic metabolism. Thus, the role(s) of LD allocation in the embryonic body plan remain to be characterized.

Asymmetric inheritance of LDs is observed across widely divergent species. For example, in mouse [4,7], LDs become motile after fertilization [7] and just prior to implantation are present at different densities in the embryo proper vs the trophoblast (zygote-derived support cells) [4]. In Xenopus [3] and zebrafish [8], LDs are deposited into the yolk sac. Over the course of embryogenesis, zebrafish embryos then transport these stored lipids to the periphery, first by using acto-myosin to transport entire LDs to neighboring cells that remain connected to the yolk via intracellular bridges [8], then in older embryos by packaging LD-derived lipids into lipoprotein particles. These particles are secreted into the bloodstream and reach distant tissues via the vasculature [9]. This pattern of unequal allocation is not unique to deuterostomes, but also occurs among species belonging to the other two major clades of Bilateria, Ecdysozoa and Spiralia. During cellularization in Drosophila embryos (an Ecdysozoan), LDs are allocated predominately to the incipient peripheral epithelium, while they are depleted from the interior yolk cell [10,11]. In mud snails (*Tritia* formerly *Ilyanassa*, a Spiralian), LDs are differentially allocated by the 4-Cell Stage (Fig 1B): the cells at the animal pole receive the LDs, while the cell at the vegetal pole receives most of the yolk protein vesicles (consistent with LD distribution in the zygote [12]). Uneven distribution of LDs during embryogenesis has even been reported in a demosponge (Poriferan) [13], where LDs are inherited by the ciliated lineage. Thus, major metazoan embryo model systems display lineage specific LD inheritance (Fig 1A).

The mechanism of unequal allocation is best understood in Drosophila. Here, the embryo develops initially as a syncytium progressing through several synchronous rounds of nuclear

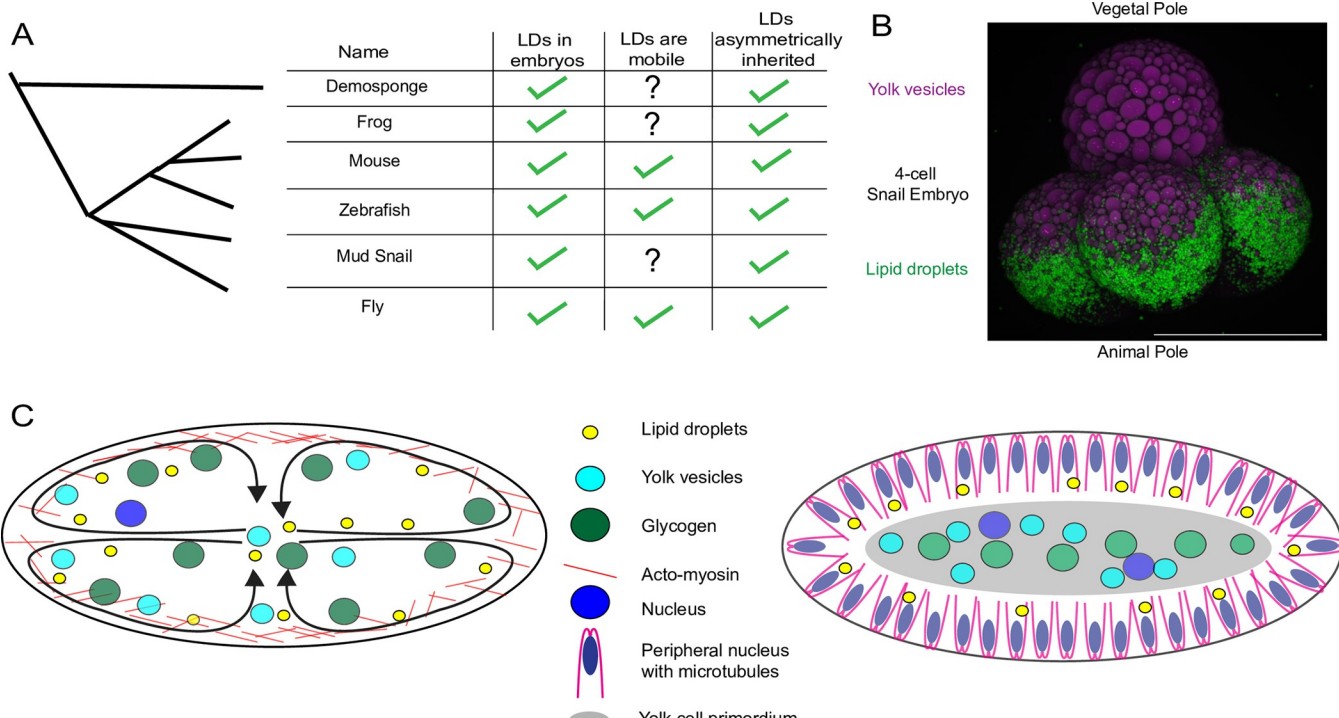

**Fig 1. Metazoan embryos asymmetrically allocate their LDs early in embryogenesis.** A) A cartooned phylogeny of the metazoan embryonic systems where LDs have been shown to be asymmetrically inherited early in embryogenesis. The left column indicates that LDs are present at laying/ovulation in the species. The right column indicates whether LDs are asymmetrically inherited in the system. The center column indicates whether the LDs are also known to be moving along the cytoskeleton. Citations are Demosponge (*Mycale laevis*) [13], Frog (*Xenopus laevis*) [3], Mouse (*Mus musculus*) [4,7], zebrafish (*Danio rerio*) [8], fly (*Drosophila melanogaster*) [10], and the mud snail data is from this paper and shown in panel B. B) *Tritia obsolete* (an ocean snail with the common name 'mud snail') embryo at the 4-cell Stage with the vegetal pole at the top and animal pole at the bottom. LDs are labeled with BODIPY493/503 in green and yolk-protein vesicles are imaged with intrinsic autofluorescence in magenta. Note that only the cells at the animal pole have inherited LDs. Scale bar 100 μm. C) Cartoons of the modes of lipid droplet motion in syncytial Drosophila embryos. On the left, a syncytial-cleavage stage embryo (Stage 1–3) undergoes bulk cytoplasmic streaming; actomyosin driven contractions at the embryonic cortex circulate the contents including LDs and other nutrient storage structures. On the right, a stage 5 embryos undergoes a bulk cytokinetic event; LDs interact with the peripheral microtubules and are transported along them into the forming cells. Note that this separates LDs (in the periphery) away from the centrally localized glycogen and yolk-protein vesicles in the embryonic center/yolk cell.

division. The embryo then undergoes a bulk cytokinetic event (cellularization), which generates a peripheral epithelium and an interior yolk cell. During this period, LDs redistribute from a homogeneous distribution to a peripheral enrichment. This redistribution depends first on bulk flow [11] as actomyosin at the cortex circulates the embryo's cytoplasm in synchrony with the nuclear divisions (Fig 1C, left). Second, before and during cellularization, LDs move bidirectionally along peripheral microtubules, via a motor complex that includes cytoplasmic dynein and kinesin-1(Fig 1C, right) [14,15,16,17,18]. This LD transport ensures that most LDs are deposited into the peripheral cell lineages [10]. While in other animal species the mechanisms of asymmetric LD inheritance remain unknown, it seems likely that cytoskeletal-mediated transport plays a crucial role, as LD motility has also been observed in mouse [7] and zebrafish [8] embryos.

To investigate why LDs in Drosophila embryos are predominately allocated to the peripheral epithelium, we disrupted LD transport by genetically abolishing two LD proteins, Jabba [11] and Klar [17]. In the absence of Jabba, LDs physically interact with glycogen granules and are inappropriately sorted with them to the yolk cell [11]. In the absence of the motor cofactor Klar, the relative activities of kinesin-1 and cytoplasmic dynein on LDs are altered, leading to LDs being transported to the yolk cell. Although protein null alleles of Jabba and Klar lead to

misallocation of LDs from the peripheral epithelium to the yolk cell, they do so by different mechanisms, and thus we expect that shared organismal phenotypes of these mutants are attributable to their shared output of LD allocation to the wrong lineage (the yolk cell).

We find that compared to wild-type embryos both mutants have slower and incomplete LD consumption and show significantly delayed embryo hatching. We observe a similar hatching delay in embryos mutant for the LD protein dPlin2. Because *dPlin2^-/-* embryos have normal LD distribution but reduced LD numbers, we conclude that the hatching delay results from reduced access to LDs in the peripheral tissues. Global gene expression analysis indeed identifies a shared set of genes altered in all three mutants, suggesting that embryos actively respond to LD deprivation. These genes span a wide range of categories, including sugar and lipid metabolism, developmental signaling pathways, and glutathione metabolism. Zygotic RNAi targeting the lipase ATGL/Bmm, Glutathione synthase, and Glutathione S-transferase T4 reduce viability in LD mutant backgrounds but not in wild type. Further, *Jabba^-/-*, *klar^-/-*, and *dPLIN2^-/-* embryos all have significantly more peroxidated lipids than wild-type counterparts. We conclude that asymmetric lineage inheritance of LDs early in embryogenesis bolsters subsequent utilization by those lineages, protecting against oxidative stress and starvation.

## Results

### Jabba and Klar ensure LD transport through two disparate means

Drosophila mothers provide their embryos with neutral lipids stored in LDs, and carbohydrates stored in glycogen granules (GGs) [19]. Under normal conditions, these storage structures do not interact with each other and are sorted into two distinct tissues during cellularization [11], LDs into the peripheral cells and GGs into the interior yolk cell. Previous work revealed that mutations in the LD proteins Klar and Jabba lead to inappropriate deposition of most LDs into the yolk cell. In the absence of the motor cofactor Klar, the movement of individual LDs along microtubules is dramatically reduced and net transport is inward, towards the plus ends of microtubules [10,16,17]; a similar phenotype is observed with hypomorphic dynein alleles [10,16,17]. In the absence of Jabba, LDs stick inappropriately to the surface of GGs, generating enormous composite structures that drag the LDs into the embryonic interior [11].

To determine if these two mutations indeed disrupt LD allocation in distinct ways, we tested for LD and GG association in *klar* mutants (allele referred to as *klar^-/-*, see methods for details) and tested LD movement in *Jabba* mutants. We first co-labeled LDs and GGs in fixed embryos, using the neutral lipid dye BODIPY and fluorescent Periodic Acid Schiff (fPAS) staining, respectively. *Jabba^-/-* embryos displayed a dramatic redistribution of LDs to the surface of GGs, forming LD rings on the surface of GGs (Fig 2B). No such association was present in the *klar^-/-* and wild-type embryos (Fig 2B), confirming previous results [11]. Next, we co-microinjected dyes to label LDs and microtubules into living embryos [20] and monitored LD motion during Stage 5 (cellularization) through timelapse imaging (S1–S3 Videos). In wild-type embryos, LDs were highly mobile and switched between long stretches of anterograde kinesin-1-mediated motion (away from the centrosome) and retrograde, dynein-mediated motion (towards the centrosome) (S1 Video). In the *klar^-/-* embryos, LDs were slower and covered much shorter distances, as previously observed (S3 Video) [17]. In *Jabba^-/-*, we are imaging the small minority of the LDs (~8% of the total) not bound to GGs. This LD population was highly mobile, like in wild type (S2 Video). Further, the gross LD distribution in the embryo periphery was different (Fig 2C). Wild type has a large population of LDs located basal to the nuclei (i.e., towards the embryo center) and a smaller, mobile population at the level of

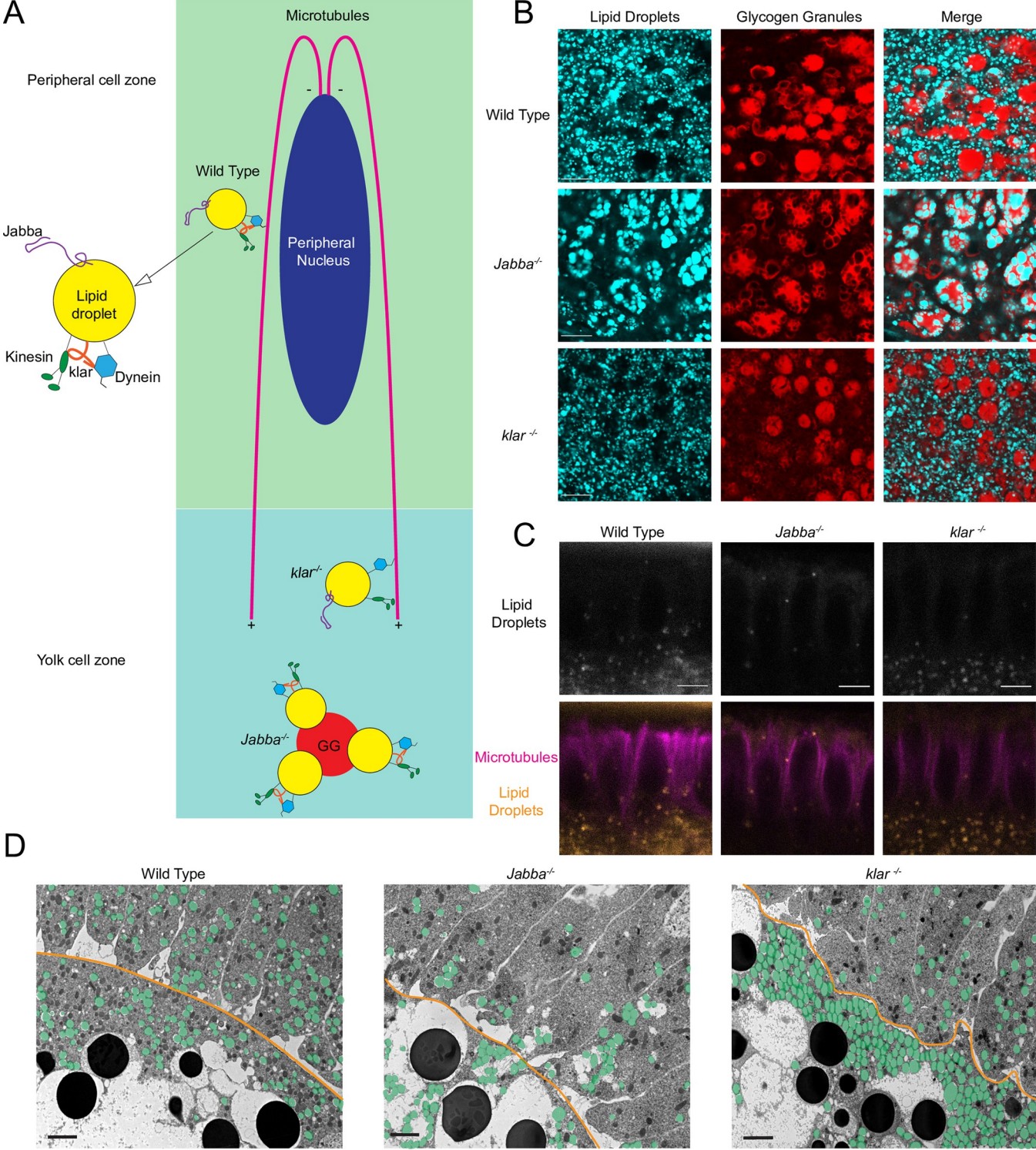

**Fig 2. Jabba and Klar are important for peripheral LD allocation through two separate means.** A) Model for how wild-type, *klar*⁻/⁻, and *Jabba*⁻/⁻ LDs interact with peripheral microtubules during cellularization. The wild-type LD moves towards the minus end of the microtubule, taking it to the periphery. The *klar*⁻/⁻ LD can bind microtubules, but cannot move properly once bound, leaving it stranded at the plus end in the presumptive yolk cell. The *Jabba*⁻/⁻ LDs are entrapped in glycogen granules and dragged into the embryonic interior with the glycogen. B) Newly laid wild type, *Jabba*⁻/⁻, and *klar*⁻/⁻ embryos stained with BODIPY493/503 (cyan) to label LDs and fPAS (red) to label glycogen granules. Note the strong association of LDs with glycogen granules in *Jabba*⁻/⁻, but not wild type or *klar*⁻/⁻. Scale bars are 10 μm. C) Frames of movies from live wild type, *Jabba*⁻/⁻, and *klar*⁻/⁻ embryos at cellularization, with LDs labeled with BODIPY493/503 and microtubules labeled with SPY-tubulin. The field-of-view has ~5 blastoderm nuclei in the center of the image with the eggshell at the top

and incipient yolk cell out of frame at the bottom of the image. Note that the *Jabba*^-/- genotype's LD distribution is different from both wild type and *klar*^-/-. Further, the few *Jabba*^-/- LDs that are free from glycogen have no issue migrating to the minus end of microtubules. Scale bars are 5 μm. D) TEMs of wild type, *Jabba*^-/-, and *klar*^-/- embryos after completion of cellularization. LDs are false colored in green. Scale bars are 2 μm. The area where the blastoderm cells meets the yolk cell is shown outlined in orange. Note that this view shows only the bottom ~1/6th of the peripheral cells but all the yolk cell cytoplasm that contains LDs (excluding *Jabba*^-/- where LDs are present much deeper in the yolk cell). Note that LD distributions are different in all three genotypes, further supporting different means of arriving at the phenotypes, but that both *Jabba*^-/- and *klar*^-/- have an excess of LDs in the yolk cell.

the nuclei or apical to them (i.e., towards the plasma membrane). In *klar*^-/-, we see mostly the former population and in *Jabba*^-/- the latter.

We also examined LD distribution in newly cellularized embryos by transmission electron microscopy (TEM), focusing on the boundary between peripheral cells and the interior yolk cell (Fig 2D). In wild type, the density of LDs was comparable in the peripheral epithelium and the outskirts of the yolk cell. In *Jabba* mutants, the LD density in the yolk cell was much higher than in the periphery (~4x), with LDs not only in the peripheral yolk-cell cytoplasm, but also occupying the more central areas typically occupied only by glycogen and yolk vesicles (Fig 2D) [11]. In *klar*^-/- mutants, there is roughly 3x the LD density in the yolk cell, with most of those LDs occupying the peripheral cytoplasm seemingly queued just beneath the completed cellularization boundary (Fig 2D, boundary in orange).

Thus, even though *Jabba* and *klar* mutants both display misallocation of LDs to the yolk cell, they differ in LD-GG association, LD motility, and the detailed spatial distribution of LDs in the yolk cell. We therefore conclude that they misallocate LDs through different mechanisms.

### *Jabba*^-/- and *klar*^-/- mutants are delayed in consuming their LDs

To address the fate of the mislocalized LDs in *Jabba* and *klar*^-/- mutants, we performed neutral-lipid thin layer chromatography (TLC) on extracts from newly laid and Stage 16 embryos (Fig 3A). In the newly laid embryos, the levels of all lipid classes detected were comparable: sterol esters (LDs) and TAG (LDs) as well as those of membrane sterols and eggshell waxes (Fig 3A). At the later timepoint, TAG levels were reduced in all genotypes, consistent with its previously described developmental turnover [5,6]. However, TAG levels were significantly higher in *Jabba*^-/- and *klar*^-/- (using membrane sterols as a reference) compared to wild type while the other lipid species were more comparable (Fig 3A). This result indicates that turnover of the LD-stored TAG is specifically compromised in the mutants.

To determine the fate of the mis-allocated LDs, we performed whole-mount neutral lipid staining which revealed abundant signal in Stage 16 *Jabba*^-/- and *klar*^-/- embryos, agreeing with our previously shown LD persistence in newly hatched *Jabba*^-/- L1 larva [11]; in contrast, wild type embryos showed barely any signal throughout (Fig 3B). Thus, the embryos start with similar LD levels, but LDs mis-allocated to the yolk cell persist in *Jabba*^-/- and *klar*^-/- embryos.

As an independent approach, we measured TAG levels biochemically. Embryos of three different ages were lysed, and TAG was detected enzymatically (Fig 3C). In the first 90 minutes of embryogenesis (Stages 1–3), wild type, *Jabba*^-/- and *klar*^-/- had similar levels of TAG, showing these genotypes were laid with comparable amounts of TAG. At 8 hours (Stages 8–10), TAG levels were still similar, though wild type started to pull ahead in TAG consumption (Fig 3C). Finally, at ~15 hours into development (Stage 16), wild-type TAG levels had dipped to the limit of detection, unlike for *Jabba*^-/- and *klar*^-/- mutants (Fig 3C). The pattern in the wild type is consistent with the previously described developmental turnover of triglycerides [5,6]. As lack of Jabba and Klar causes LD misallocation by different means, it seems likely that it is the misallocation itself that results in compromised LD turnover.

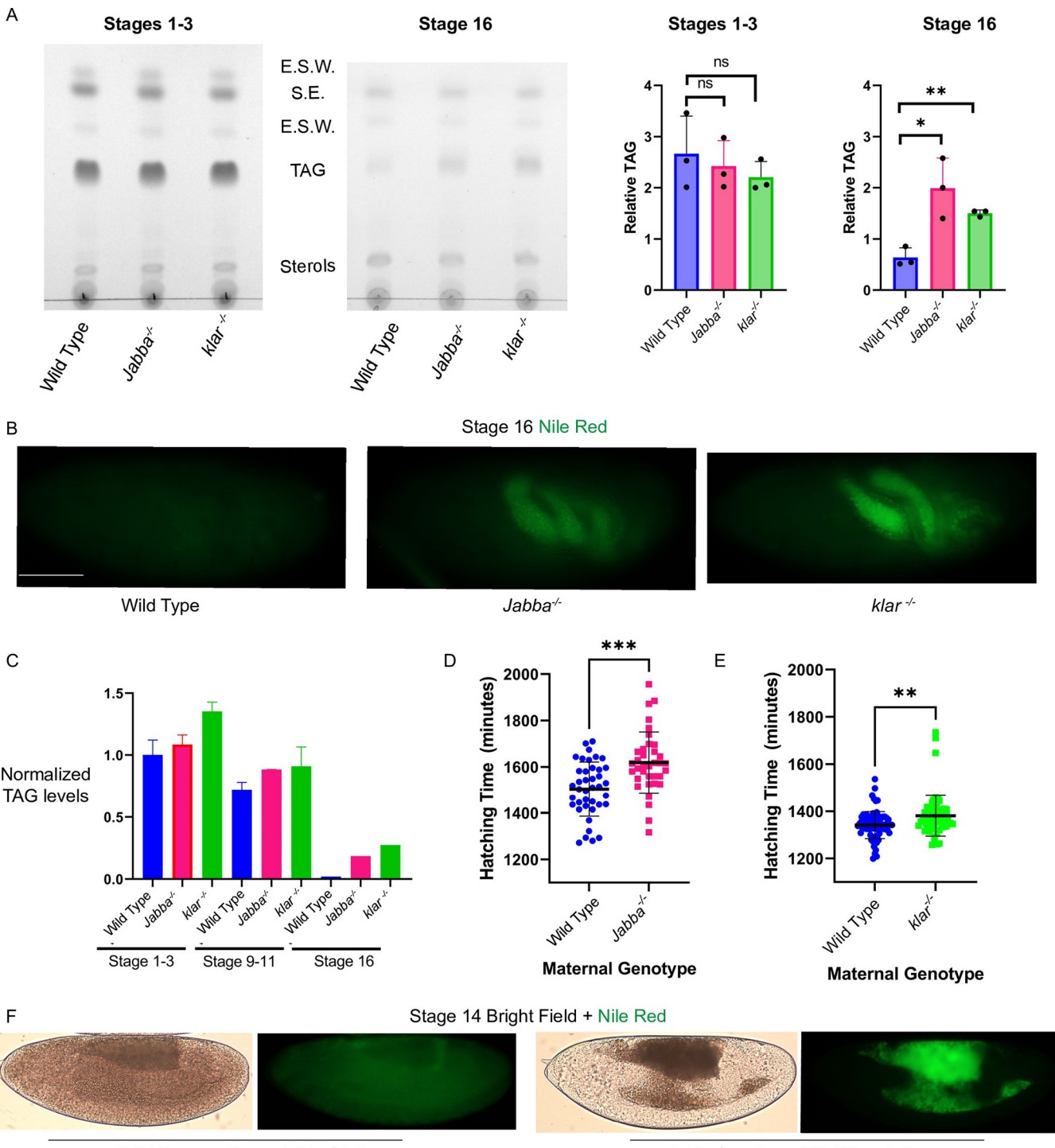

**Fig 3. In *Jabba⁻ᐟ⁻* and *klar⁻ᐟ⁻* embryos, misallocated LDs are not consumed appropriately, and embryogenesis is protracted.** A) Neutral lipid TLCs of wild type, *Jabba⁻ᐟ⁻*, and *klar⁻ᐟ⁻* embryos at a young (prior to the syncytial blastoderm) and older stage (~2/3rds through embryogenesis). Quantification of the LD-stored TAG, controlled against membrane sterols and protein concentration, shows embryos start with similar levels. However, *Jabba⁻ᐟ⁻* and *klar⁻ᐟ⁻* have significant TAG persistence at Stage 16. E.S.W. is eggshell wax. S.E. is sterol esters. TAG is triglycerides. N is three replicates of 100 embryos each per genotype. B) Nile Red whole mount staining of Stage 16 embryos shows the persistent LD population in *Jabba⁻ᐟ⁻* and *klar⁻ᐟ⁻* in the yolk cell. Scale bar 100 μm. C) Enzymatic TAG measurements for wild type, *Jabba⁻ᐟ⁻*, and *klar⁻ᐟ⁻* embryos at an early, mid, and late embryogenesis timepoint. TAG persists in *Jabba⁻ᐟ⁻*, and *klar⁻ᐟ⁻* embryos. TAG measurements were first normalized to protein levels in the sample, and then to the average reading from the Stage 1–3 wild-type samples. N is three replicates of 100 embryos each per genotype. D) Hatching assay data for a reciprocal cross setup for wild type and *Jabba⁻ᐟ⁻* genotypes. Blue is

wild-type mothers x *Jabba*<sup></sup> fathers and magenta is *Jabba*<sup>-/-</sup> mothers x wild-type fathers. E) Reciprocal cross setup for wild type and *klar*<sup>-/-</sup> genotypes. Blue is wild-type mothers x *klar*<sup>-/-</sup> fathers and green is *klar*<sup>-/-</sup> mothers x wild-type fathers. D,E) The hatching assays were performed at 22˚C for *Jabba*<sup>-/-</sup> and 25˚C for *klar*<sup>-/-</sup>; *Jabba*<sup>-/-</sup> was done at a lower temperature as embryonic viability in this genotype drops at 25˚ C [31]. There are at least 40 embryos per genotype. F) Stage 14 embryos stained with Nile Red and imaged from the crosses used in D, demonstrating that the *Jabba*<sup>-/-</sup> LD misallocation is a result of the maternal genotype. A,D) Significance was determined with unpaired Student's t-tests in GraphPad Prism, ns is P > 0.05, * is P ≤ 0.05, ** is P ≤ 0.01, *** is P ≤ 0.001. Error bars represent standard deviation.

## Embryogenesis is protracted in *Jabba*<sup>-/-</sup> and *klar*<sup>-/-</sup> mutants

Because embryos mutant for either Jabba or Klar can give rise to fertile adults [17,21], proper turnover of the embryo's entire LD population is apparently not required for viability. However, disruption of embryogenesis (e.g., by manipulating environmental parameters such as temperature and oxygen levels) can dramatically affect the duration of embryogenesis without reducing hatching success [22,23]. We therefore assessed how long embryos of various genotypes take until they hatch into larvae, as a more sensitive measure of compromised embryogenesis. Control- and experimental-genotype embryos were collected in parallel for ~1hr and visually confirmed to be in the appropriate stage (e.g., prior to cellularization); then embryos from each genotype were transferred onto the same assay plate and arranged into rows so that they fit into the field of view of our camera setup. The plates with embryos were aged overnight and then videorecorded to observe hatching events.

Initial experiments suggested that embryos from strains mutant for Jabba or Klar display hatching delays. Mislocalization of LDs in embryos is due to the lack of Klar [17] or Jabba (Fig 3F) in the mother, yet Jabba and Klar are expressed both during oogenesis and embryogenesis [24] (FlyBase, see Materials and Methods for citation details). It is therefore possible that altered timing of embryogenesis results from new expression of these proteins in the zygote and would thus be unrelated to LD mislocalization. For example, it is known that proper development of the embryonic salivary gland requires zygotically expressed Klar [25]. To circumvent this issue, we compared hatching times between embryos from wild-type mothers and LD mutant fathers (*Jabba*<sup>-/-</sup> or *klar*<sup>-/-</sup>) to embryos from reciprocal crosses. In this comparison, the embryos have the same zygotic genotype with respect to *Jabba* and *klar* but differ in LD allocation. The reciprocal crosses were performed at different temperatures, as *Jabba*<sup>-/-</sup> viability drops at 25˚C. Intriguingly, embryos from LD mutant mothers took significantly longer to hatch than their counterparts from wild-type mothers (Fig 3D for *Jabba*<sup>-/-</sup>, and 3E for *klar*<sup>-/-</sup>), with embryos from *Jabba*<sup>-/-</sup> mutant mothers taking 90mins longer to hatch (Fig 3C), and embryos from *klar*<sup>-/-</sup> mothers taking 40mins longer (Fig 3E) to hatch than their relevant controls. Thus, embryos which mislocalize their LDs and fail to fully consume them display prolonged embryogenesis.

## Embryos with diminished LD numbers are also delayed in hatching

Our results from the two preceding sections suggest that the hatching delay is caused either by too few LDs in the periphery or too many LDs in the yolk cell. For example, too few LDs might create a 'starvation' state in peripheral cells or yolk-cell lipid overload might result in metabolic stress. To distinguish between these possibilities, we analyzed mutants with reduced maternal LD loading. Embryos lacking the LD-resident perilipin protein dPLIN2/LSD-2 receive less triglyceride from their mothers [26] but contain LDs of normal size [27], suggesting that they contain fewer LDs. Indeed, neutral lipid staining (BODIPY) in newly laid embryos reveals lower LD density in *dPLIN2*<sup>-/-</sup> embryos (Fig 4C). We then stained wild type, *Jabba*<sup>-/-</sup>, *klar*<sup>-/-</sup>, and *dPLIN2*<sup>-/-</sup> embryos for neutral lipids and compared them at various developmental stages (Fig 4A). In newly laid embryos, LDs were homogeneously distributed in wild type, *klar*<sup>-/-</sup>, and

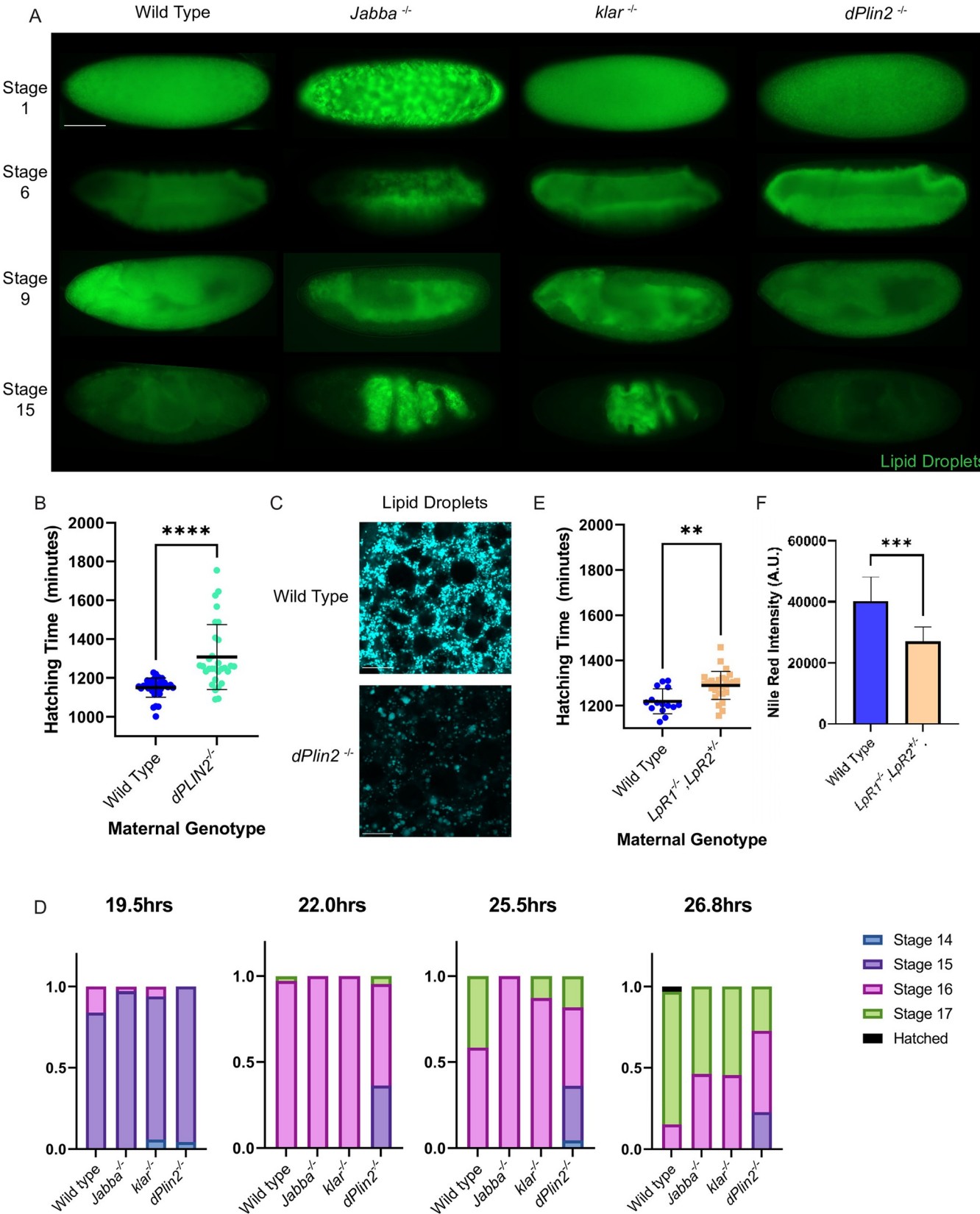

**Fig 4. Mutants with defective maternal LD loading have hatching delays.** A) Whole mount, Nile Red stained embryos (LD-mutants and wild type) at different timepoints. *dPLIN2*⁻/⁻ embryos appropriately localize a smaller pool of LDs. Scale bar 100 μm. B) Hatching assay for a reciprocal cross setup for wild type and *dPLIN2*⁻/⁻ genotypes. Blue is wild-type mothers x *dPLIN2*⁻/⁻ fathers and turquoise is *dPLIN2*⁻/⁻ mothers x wild-type fathers. *dPLIN2*⁻/⁻ is delayed like *Jabba*⁻/⁻ and *klar*⁻/⁻. C) Newly laid,fixed-embryo, LD-staining with BODIPY. Scale bar 5 μm D) Shows the stage progression of embryos synchronized at Stage 14/dorsal closure. Note the slight onset of a delay, suggesting that the bulk of the hatching delay seen in mutants occurs during Stage 17/hatching/when the embryo is ramming its head into the eggshell. E) Hatching assay using a reciprocal cross setup for wild type and *LPR1*⁻/⁻,*LPR2*⁺/⁻ genotypes. Blue is wild-type mothers x *LPR1*⁻/⁻ *LPR2*⁺/⁻ fathers and beige is *LPR1*⁻/⁻ *LPR2*⁺/⁻ mothers x wild-type fathers. Note that 1 copy of LPR2 is required for completion of oogenesis, making the zygotic genotypes heterogeneous in this cross. Embryos from *LPR1*⁻/⁻ *LPR2*⁺/⁻ mothers are delayed like *Jabba*⁻/⁻ and *klar*⁻/⁻ *dPLIN2*⁻/⁻. B,D,E) Each genotype starts with at least 25 embryos. F) Stage 1–4 embryos of the respective genotype stained with Nile red and quantified using FIJI. N is 14 embryos for wild type and nine for *LPR1*⁻/⁻ *LPR2*⁺/⁻. B,E,F) Significance was determined with unpaired Student's t-tests in GraphPad Prism, ns is P > 0.05, * is P ≤ 0.05, ** is P ≤ 0.01, *** is P ≤ 0.001, and **** is P ≤ 0.0001. Error bars represent standard deviation.

*dPLIN2*⁻/⁻, while in *Jabba*⁻/⁻ LDs displayed a patchy distribution, consistent with redistribution to the surface of GGs (Fig 4A). At Stage 6, wild type and *dPLIN2*⁻/⁻ embryos' LD localization remained indistinguishable, while in *Jabba*⁻/⁻ and *klar*⁻/⁻ embryos LDs were mostly localized to the yolk cell (Fig 4A). In Stages 9 and 15, wild-type and *dPLIN2*⁻/⁻ embryos displayed the same global LD distribution, with overall diminished staining intensity in *dPLIN2*⁻/⁻ compared to similarly aged wild-type embryos (Fig 4A). At the same stages, *Jabba*⁻/⁻ and *klar*⁻/⁻ LDs were predominantly localized to the yolk cell, with minor signal in the periphery in Stage 9, but not Stage 15, suggesting that the appropriately localized LD population is quickly consumed while the yolk cell LDs are retained (Fig 4A). Thus, *dPLIN2*⁻/⁻ mutants appropriately localize a smaller pool of LDs, i.e., they represent embryos with fewer LDs in the periphery, but without an excess of LDs in the yolk cell.

We performed reciprocal crosses of *dPLIN2*⁻/⁻ and wild-type adults, and assayed the time taken for the progeny to hatch (Fig 4B). The embryos of *dPLIN2*⁻/⁻ mothers took on average 120min longer to hatch than embryos from wild-type mothers. Closer inspection of the hatching data revealed an expanded tail for lipid-deprived embryos (*Jabba*⁻/⁻, *klar*⁻/⁻, or *dPLIN2*⁻/⁻): the 4ᵗʰ quartile hatchers were very delayed, occasionally exceeding 30hrs (~40% longer than would be expected at 25°C [22]). To understand at which stage the hatching delay occurs, we collected embryos laid during a 50-minute window and allowed them to develop for 17hrs at 22°C and then embryos at Stage 14 were transferred to a new plate. This is several hours after the proposed onset of TAG metabolism [28]. Embryos were then examined at four timepoints for their progression through the remaining stages, as judged by morphological criteria, (Fig 4D). At 19.5hrs post laying, there were no obvious differences between wild-type, *Jabba*⁻/⁻, *klar*⁻/⁻, and *dPLIN2*⁻/⁻ embryos. The wild-type embryos pulled slightly ahead through the 22hr, 24.5hr, and 26.8hr timepoints (Fig 4D). The modest differences in progressing from Stage 15 to 16, and 16 to 17 do not fully explain the observed hatching delays, suggesting that most of the delay occurs at the Stage 17 to hatching transition. This is the stage when the embryo begins active skeletal muscle movements that generate coordinated, whole-body contractions that eventually break the eggshell. These contractions start at the posterior, move anteriorly, and end with the mouthparts engaging with the eggshell. All three LD-deprived embryos spend more time in this 'hatching contraction' period. S4 Video shows a reciprocal cross between wild type and *klar*⁻/⁻ wherein two out of four *klar*⁻/⁻ embryos spend several hours (5+) more trying to hatch than counterparts from wild-type mothers. This observation is also true for embryos from *Jabba*⁻/⁻ and *dPLIN2*⁻/⁻, with *dPLIN2*⁻/⁻ being particularly severe (~1/3 of embryos attempt but fail to hatch). Thus, loss of Jabba, Klar, or dPLIN2 seems to converge on the same developmental responses, likely in Stage 17 (Fig 4D).

Finally, we used mothers with reduced levels of the lipoprotein receptors LpR1 and LpR2. During oogenesis, these receptors facilitate the uptake of extracellular lipids which are used to generate the vast majority of the LDs present in oocytes [29]. Thus, LpR1 and LpR2 are not LD proteins themselves, but allow us to manipulate how many LDs embryos inherit. Complete loss

of the LpRs derails oogenesis, so we combined an allele lacking both *LpR1* and *LpR2* [29] in trans with one lacking *LpR1* [29] (i.e., 0x *LpR1* and 1x *LpR2*). This 0x *LpR1* and 1x *LpR2* genotype has ~70% of LDs deposited into wild type embryos [29]. We then reciprocally crossed males and females of this genotype to wild type and performed a hatching assay. In these crosses not all embryos have the same zygotic genotype because one parent is not homozygous, but the reciprocal crosses generate embryos of the same genotypes with respect to *LpR1* and *LpR2*. The embryos from the LD-deprived *LpR1* and *LpR2* deficient mothers were delayed relative to the controls, by 70 minutes (Fig 4E). Thus, embryos with reduced maternal LD loading take longer to hatch than embryos from wild-type mothers, like embryos with mislocalized LDs (*Jabba*[-/-] *and klar*[-/-]). These results show four independent LD mutations converge on a developmental delay.

## A core set of genes responds to loss of access to LDs

We next sought to determine if the embryo's lipid-deprived state results in changes to its transcriptome. It is conceivable that lack of maternal Klar, Jabba, or dPLIN2 affect embryonic gene expression by multiple mechanisms in addition to lipid deprivation; for example, Klar is known to be important for proper localization of the posterior determinant Oskar in the early embryo [30], and Jabba controls nuclear levels of the histone H2Av before cellularization [31]. However, we reason that transcriptional responses shared between embryos from mothers mutant for Klar, Jabba, or dPLIN2 are most likely due to lipid deprivation, especially since mutations in the three proteins lead to such deprivation by distinct mechanisms.

We therefore isolated total RNA and enriched for mRNA from Stage 15 embryos (phenotypes cartooned in Fig 5A), which is roughly 2/3s of the way through embryogenesis (shown in Fig 4A) and several hours post the reported onset of fat consumption [28]. The embryos were collected from four maternal genotypes (wild type, *klar*[-/-], *Jabba*[-/-], or *dPLIN2*[-/-]), crossed to the same paternal genotype (wild-type, OrR). Thus, all these embryos shared about half of their zygotic genome, with the varying lipid deprivation phenotypes resulting from the maternal genotypes.

We then analyzed the global transcriptome by RNA-sequencing. Triplicate measures of each genotype were assembled into estimated read counts; then each mutant was independently compared to wild type to identify those genes that are differentially expressed (significantly higher or lower levels of RNA were found in the mutants relative to wild type). We found 810 such genes in *Jabba*[-/-], 420 in *klar*[-/-], and 620 in *dPLIN2*[-/-] (Fig 5E and S1, S2, S3, S4, S5, S6, S7, and S8 Tables). Remarkably, 110 genes changed in the same manner (either up or down) in all 3 mutants (Fig 5C); these genes thus likely represent a core set whose expression is altered in response to LD duress (Fig 5B). Our findings suggest that lipid-deprived embryos mount an active transcriptional response.

These genes fit into a diverse set of processes and to make them more accessible we subjectively categorized them based on their functional annotations (FlyBase, Fig 5D). The categories included oxidative stress, mitochondrial function, metabolism, sugar responsive, transcriptional regulators, membrane transporters, proteostasis, and a 'other' category that includes many genes involved in mitosis and lineage acquisition. Fig 5F shows individual IGV tracks for a gene of interest, *ATGL/Bmm*, which is significantly upregulated in *Jabba*[-/-] and *dPLIN2*[-/-]. Intriguingly, we identified 68 genes consistently upregulated in all the mutant conditions (candidates of interest shown with green symbols Fig 5E), which might suggest that they are beneficial for managing the LD deprived state.

## LD-deprived embryos have large alterations to their metabolic proteome

Uncovering the biological consequences of this transcriptional response is complicated by the fact that the mutant embryos also display dramatic changes in mRNAs for genes involved in

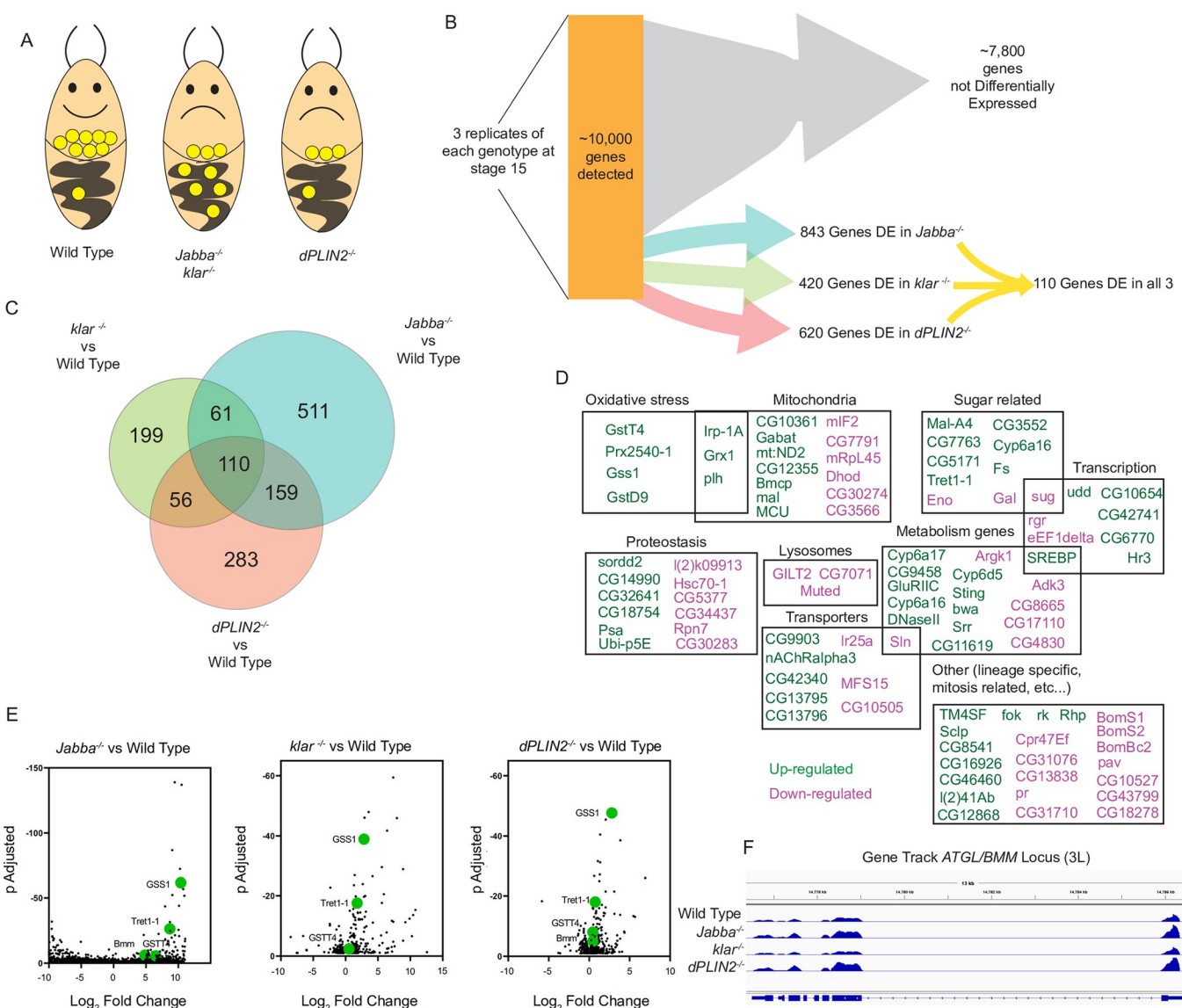

**Fig 5. LD-deprived embryos have a shared transcriptional response.** A) Cartoon showing the wild-type LD distribution, the LD-transport mutant distribution, and the maternal loading mutants' distribution. B) The number of genes detected in our mRNAseq analysis and the numbers of differentially expressed genes in the three mutants. N is three replicates of three embryos per genotype. C) A Venn diagram showing shared differentially expressed genes in each of the three mutants. D) Subjective binning of each of the 110 shared differentially expressed, protein-coding genes based on the FlyBase annotations. The color coding is done by whether it is upregulated (green) or downregulated (magenta) in all three mutants. E) Volcano plots for the differentially expressed genes in the three mutants' comparisons. Genes which are investigated in later figures are shown with enlarged, green symbols and labeled. Note that the scale of the y-axis for the *Jabba*[-/-] comparison's is different to fit the data. F) IGV tracks for 1 replicate of wild type, *Jabba*[-/-], *klar*[-/-], and *dPLIN2*[-/-] showing expression levels of the ATGL/Bmm locus on the 3[rd] chromosome.

proteostasis. Alterations to protein turnover pathways might lead to additional changes not captured in the RNA seq data or counteract the observed mRNA level changes. Prior to conducting a proteome-wide analysis (Fig 6), we took advantage of the fact that for one of our candidates, the trehalose transporter Tret 1–1, an antibody exists that has been verified for immunostaining [32]. Tret 1–1 is an essential sugar transporter in Drosophila [32,33], and our RNA-seq data show upregulation in our three mutants (Fig 7A). We therefore fixed embryos and stained them with an antibody against Tret 1–1 (targeting the PA isoform). To assess

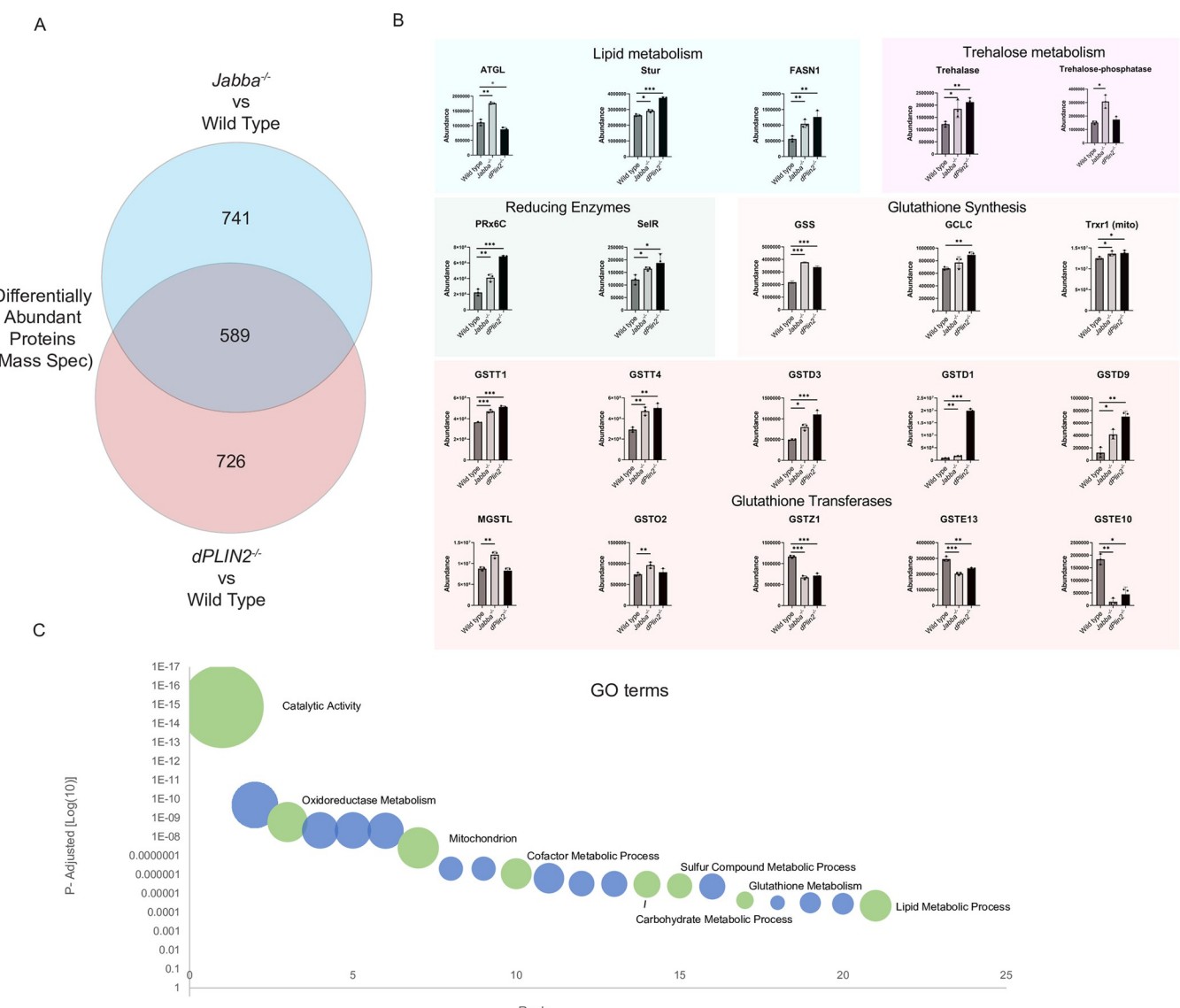

**Fig 6. LD deprived embryos (*Jabba^-/-* and *dPLIN2^-/-* mutants) have changes in protein levels for many proteins, including those involved in lipid metabolism, trehalose metabolism, and redox metabolism.** A) Venn diagram showing overlapping and nonoverlapping differentially expressed genes in the *Jabba^-/-* vs wild type comparison compared to the *dPLIN2^-/-* vs wild type comparison, based on the mass spec analysis. B) Selected gene-products from mass spec (LC-MS-MS) analysis of Stage 15 wild type, *Jabba^-/-*, and *dPLIN2^-/-* embryos. The Y-axis shows normalized abundance reads. Background color indicates the category the genes are binned within. For ATGL, the black asterisks indicate a significant increase relative to wild type, while the grey indicates a significant decrease. C) The top 21 significant GO terms based on proteins which are differentially abundant in both *Jabba^-/-* and *dPLIN2^-/-* embryos. They are ranked in order of significance. The size is scaled to the number of genes matching the term. Blue bubbles are unlabeled while the green bubbles are labeled and referred to in the main text.

global Tret 1–1 levels, we acquired z-stacks encompassing the entirety of Stage 15 embryos and quantified the mean intensity value of the Tret 1–1 signal across the entire volume. Consistent with our RNA-seq data, there was a global increase in protein levels in both *klar^-/-* and *Jabba^-/-* relative to wild type (Fig 7C). Similar changes were observed for individual tissues: Tret 1-1-PA signal increased in *Jabba^-/-* and *klar^-/-* in the developing nerve cord and associated developing muscle in Stage 14 embryos (Fig 7B). Thus, for this candidate, upregulation of its mRNA is also reflected in a dramatic change in protein levels.

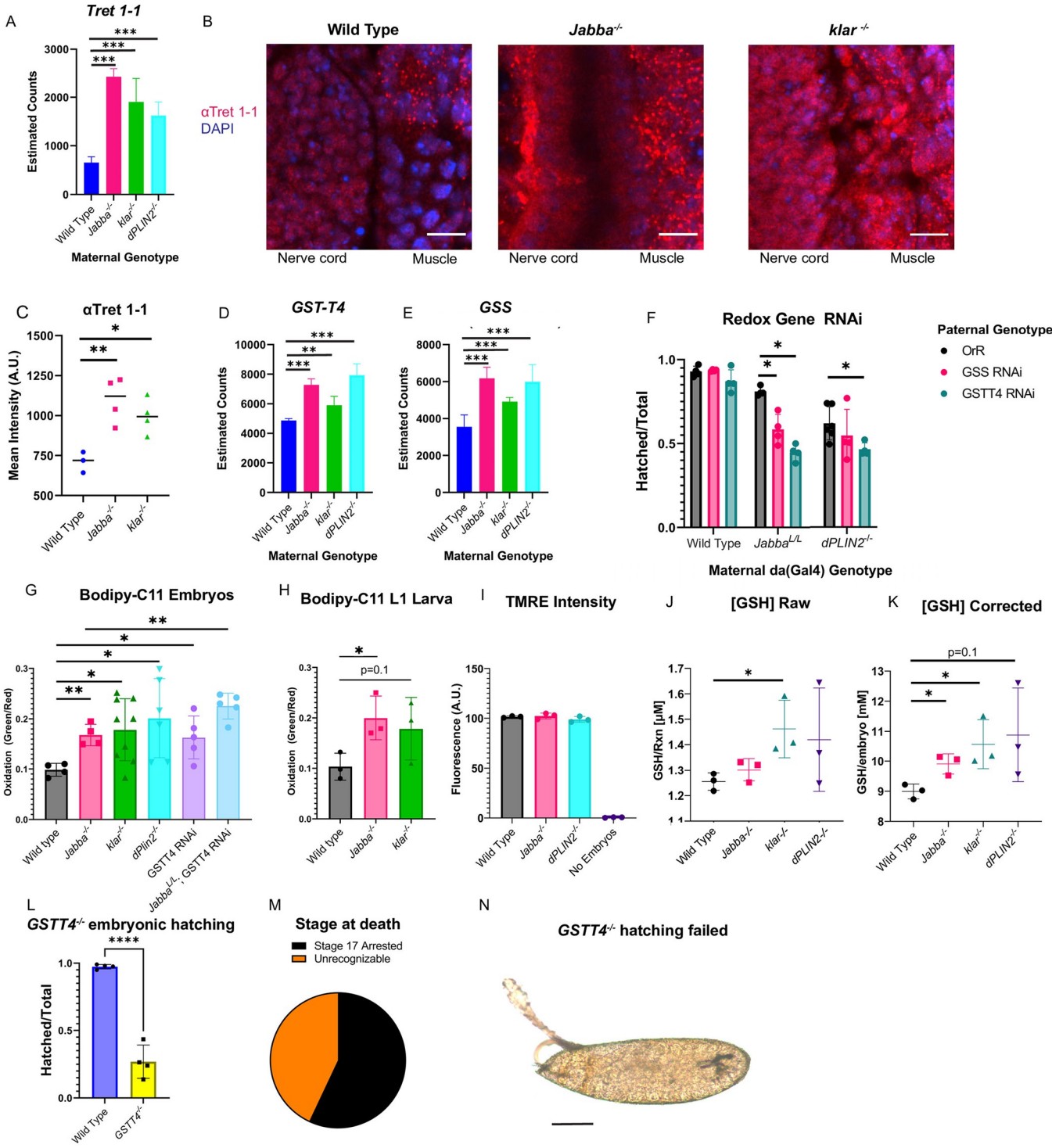

**Fig 7. LD-deprived embryos have increased levels of the sugar transporter Tret1-1 and of lipid peroxidation as well as changes to glutathione metabolism.** A) Relative levels of Trehalose Transporter 1-1/Tret1-1 mRNA in the respective genotypes (Wild type, *Jabba*⁻/⁻, *klar*⁻/⁻, *dPLIN2*⁻/⁻), based on RNA seq analysis. B) Immunofluorescence staining of Tret1-1 in Stage 14 embryos at the developing nerve cord: muscle junction. Scale bars 10 μm. C) Quantification of Tret1-1 staining intensity in z-stacks encompassing entire Stage 14 embryos. Images captured using spinning disc confocal microscopy. N is three embryos for wild type and four for *Jabba*⁻/⁻ and *klar*⁻/⁻. D) Relative levels of Glutathione Transferase-Theta 4/GSTT4 mRNA in the respective genotypes (Wild type, *Jabba*⁻/⁻, *klar*⁻/⁻, *dPLIN2*⁻/⁻) based on RNA seq analysis. E) Relative levels of Glutathione Synthase (GSS) mRNA in the respective genotypes (Wild type, *Jabba*⁻/⁻, *klar*⁻/⁻, *dPLIN2*⁻/⁻) based on RNA seq analysis. Note we are pooling reads for the recently/tandemly duplicated GSS1 and GSS2 loci [36]. F)

Zygotic RNAi (UAS) for GSTT4 and GSS (targets both GSS1 and GSS2) driven by *da-Gal4*. Note the phenotypes are mild, but a large supply of GSS protein is supplied by the mother which is unaffected by our RNAi and several other GSTT family members are expressed in the embryo (GSTT1 and GSTT2). N is roughly 100 embryos scored on four different days from at least two sets of crosses per genotype. G) Fluorescence readings from a BODIPY C11 lipid-peroxidation assay in pooled Stage 14–16 embryo homogenates. Data is shown as green (oxidized) fluorescence divided by the red (reduced) fluorescence. N is four replicates for wild type and *Jabba*[-/-], nine for *klar*, six for *dPLIN2*, and five for GSTT4 RNA and *Jabba*[L/L];GSTT4 RNAi of 10 embryos per genotype. H) Homogenates of L1 larvae which have hatched <2hrs prior assayed for BODIPY C11 peroxidation. LD-mutant L1 larva carry their oxidative burden post hatching. N is three replicates of 10 L1 larva per genotype I) TMRE was used to assess the mitochondrial inner membrane potential in Stage 14–16 embryo homogenates. N is three replicates of 10 embryos per genotype. J) GSH/glutathione was measured in Stage 14–16 embryos homogenates. The raw readings show the μM concentration within the reaction (rxn). K) Shows the measurement in panel J corrected for differences in embryonic volumes and displays the data as embryonic GSH concentrations. J&K) Y-axes do not start at 0. N is three replicates of six embryos per genotype. P values were calculated using unpaired, two tailed Student's t-tests C,F,G,H,J,K,L) Significance was determined with unpaired Student's t-tests in GraphPad Prism, ns is P > 0.05, * is P ≤ 0.05, ** is P ≤ 0.01, *** is P ≤ 0.001, and **** is P ≤ 0.0001. A,D,E) significance was determined by DEseq2's p-adjusted value using the 3 replicated per genotype. Error bars represent standard deviation.

While powerful for verifying changes in abundance for individual proteins, this immunostaining approach is not conducive to a comprehensive analysis of expression changes, in part because verified antibodies are only available for a few of our candidate genes. Therefore, we opted for a proteomic approach to simultaneously probe changes for thousands of proteins. LC-MS-MS was performed on wild type, *Jabba*[-/-] and *dPLIN2*[-/-] embryos. We were able to detect peptides from ~5,000 proteins in all three genotypes (details in Materials and Methods), 3,039 were at similar levels of normalized abundance (S9 Table). In addition, 588 were differentially abundant in both *Jabba*[-/-] and *dPLIN2*[-/-] embryos. These 588 differentially abundant proteins were compared to the 269 genes which were differentially expressed in the RNAseq data (Fig 5C, the 3-fold overlap and the *Jabba*[-/-] and *dPLIN2*[-/-] overlap); we found that 94 genes were detected in both and of those 33 genes were significantly different at both the protein and mRNA level. Note that we permitted mRNA and protein levels to vary in different directions, for example, *Dhod* mRNA is down in both mutants while protein levels are up. Within the 33 differentially expressed genes in both experiments were *Glutathione synthase (GSS)*, *Adipose triglyceride lipase ATGL/Bmm*, and *Glutathione s-transferase T4 (Gstt4)*, which will be examined in more detail in subsequent sections. Thus, both approaches point to qualitatively similar responses of the embryos. This congruence is particularly remarkable since–for technical reasons–the two approaches differed in experimental details (e.g., the zygotic genotype of the embryos). We conclude that lipid-deprived embryos indeed mount a dramatic active response that results in global changes to both transcriptome and proteome.

For the ~588 genes with different peptide abundance in both mutants (Fig 6A), we then performed gene ontology (GO) analysis, using genes with similar levels in all three genotypes (3,039) as background (S10 Table). We found a robust response of GO terms relating to metabolism, with the strongest hits including 'catalytic activity', 'small molecule metabolic process', 'oxidoreductase activity', and 'mitochondrion' (Fig 6C and S11 Table). Similar GO matches were found when analyzing the differentially expressed mRNA in *Jabba*[-/-] and *dPLIN2*[-/-] embryos (S8 Table). We conclude that lipid-deprived embryos massively alter their metabolic proteome.

### Enzymes for glutathione metabolism are altered in lipid-deprived embryos

To follow up on the omics data sets, we sought a response that was novel in its connection to LDs and that was following the same trajectory in all our LD-deprived genotypes. The following GO terms stood out: 'Glutathione metabolic process' enriched 3.3-fold (P = $2.2 \times 10^{-5}$), 'oxidoreductase activity' enriched 1.98-fold (p = $1.63 \times 10^{-9}$), and 'sulfur compound metabolic process' enriched 2.39-fold (p = $3.88 \times 10^{-6}$) (Fig 6C and S11 Table). Our RNAseq had found significant upregulation of several redox genes including *glutathione synthase* and *GSTT4*.

Thus, both the transcriptome and proteome analysis suggest that in our mutant embryos glutathione metabolism is altered.

Glutathione (GSH) is a tripeptide that serves as an intracellular reducing agent. For example, it is used to neutralize reactive oxygen radicals, in the process becoming oxidized to glutathione disulfide (GSSG), a dimer of two glutathione molecules joined via a disulfide bridge. Glutathione-S-transferase enzymes can utilize GSH as a reducing agent to reconcile unwanted oxidation on proteins, lipids, and other xenobiotics like pesticides [34]. GSH serves broad functions in metabolism and as an antioxidant. Because of its critical roles, its levels are controlled by multiple pathways: GSH is synthesized by the sequential action of two enzymes, glutamate-cysteine ligase (GCL) [35] and glutathione synthetase (GSS) [36]. In Drosophila, GSH is regenerated from GSSG by thioredoxin reductase (TRX) [37].

Our data suggest that lipid-deprived embryos display broad changes in multiple aspects of glutathione metabolism. In the glutathione synthesis pathway, the catalytic subunit of GCL (GCLC) is increased in $dPLIN2^{-/-}$, while the modifying subunit (GCLM) is down in both $Jabba^{-/-}$ and $dPLIN2^{-/-}$ at the peptide and mRNA level (Fig 6B and S2, S4, and S9 Tables). The second enzyme in the pathway, GSS, is increased in all three mutants at the mRNA and peptide level in both $Jabba^{-/-}$ and $dPLIN2^{-/-}$ (Fig 6B) as is the enzyme thioredoxin reductase (Trxr1) responsible for regenerating glutathione from the oxidized dimer (GSSG, Fig 6B). Of 24 glutathione transferases detected in our proteome, ten displayed significantly different peptide levels. Several members of GST subfamilies D and T were upregulated in both mutants, with the three downregulated GSTs belonging to subfamilies Z and E (Fig 6B), perhaps indicating specific GST family needs to combat LD-deprivation.

Increases in GSS, Trx1, and GCLC are expected to increase GSH levels; GCLM reduction is known to reduce GSH levels [35]. It is therefore not obvious how total GSH levels might be affected. We homogenized Stage 15 wild-type, $Jabba^{-/-}$, $klar^{-/-}$, and $dPLIN2^{-/-}$ embryos and measured free GSH concentrations in the homogenates using a luciferase-based assay. As embryos of the four genotypes have slightly different average volumes (wild type (OrR) 9.3nL, $Jabba^{-/-}$ 8.8nL, $klar^{-/-}$ 9.2nL, and $dPLIN2^{-/-}$ 8.7nL), we corrected the raw reads by volume to determine the GSH concentration within the embryo (Fig 7K; raw reads are shown in Fig 7J). Indeed, GSH levels are elevated in the mutants compared to wild type.

## LD-deprived embryos display reliance on the glutathione pathway

The altered regulation of the glutathione pathway in the lipid-deprived embryos suggests that it is important to combat stress from nutrient deprivation. We therefore examined the functional relevance of this pathway in our mutants by employing the Gal4-UAS system to drive dsRNAs targeted against GSS (combines the recently duplicated GSS1 and GSS2 [36]) and GSTT4. To avoid disrupting oogenesis, we opted for zygotic knockdown, crossing mothers carrying the *da-Gal4* driver to fathers with the UAS dsRNA constructs. Even though da may have sexually dimorphic expression levels, we picked this driver because it is expressed in the yolk cell, unlike actin and tubulin-based drivers (as judged from BDGP (see Materials and Methods for citation details) in situ: probe FI13819 for da, probe RE44641 for *alphaTub67c*, probe RE02927 for actin5C). For most experiments, we used a da-Gal4 insertion on the third chromosome; for *klar* experiments, we used an insertion of the same construct on the second chromosome. Introducing *da-Gal4* into a wild-type, $klar^{-/-}$, or $dPLIN2^{-/-}$ background proved unproblematic. However, in a background fully null for Jabba, the *da-Gal4* driver dramatically reduced embryo viability, for unknown reasons. We therefore generated an alternative line with reduced Jabba expression, in which the driver is paired with the hypomorphic allele $Jabba^{Low}$ (referred to as $Jabba^{L}$ in the following). Embryos from $Jabba^{L/L}$ mothers have an

intermediate LD allocation defect (*Jabba*$^{L/L}$ Fig 6C, wild type and *Jabba*$^{-/-}$ in Fig 4A). In a *Jabba*$^{L/L}$ background, Jabba protein levels are ~1/8 of what is observed in the wild type [24], and we noticed no ill effects from Gal4 expression on embryo viability.

Using a hairpin targeting GSTT4, we observed significant reduction in hatching success in the *Jabba*$^{L}$ and *dPLIN2* mutant backgrounds (down 35% in *Jabba*$^{L}$ and 15% in *dPLIN2*$^{-/-}$ relative to the no-hairpin controls), but not in wild type (Fig 7F). The strength of the effect is remarkable given that GSTT4 is only one of 24 glutathione transferases whose expression we detect in the embryo (S9 Table) and one of seven upregulated in both mutants (Fig 6 and S9 Table). Despite this redundancy, *Jabba* and *dPLIN2* mutants apparently rely on GSTT4 for survival.

It is conceivable that the strong effect of the hairpin is due to off-target effects, resulting in knockdown of not just GSTT4, but also other GSTs. We believe this is unlikely because a hairpin against a GSTD family member showed no reduction in hatching. However, even in such a scenario, our results still strongly argue that *Jabba* and *dPLIN2* mutants are uniquely reliant on the glutathione pathway. In addition, we found that a mutation of *GSTT4* already severely compromises embryonic development in a wild-type background: although a *GSTT4* allele due to an insertion of a MiMIC element ([38]; FlyBase, MI07169-TG4.1) is viable and fertile, over 70% of embryos from the homozygous stock failed to hatch (Fig 7L). The majority of the embryos reached Stage 17 (Fig 7M), began hatching contractions, then gradually ceased moving while turning a rusty hue (Fig 7N). Finally, attempts to make a stock double mutant for *Jabba* and *GSTT4* failed, with intermediate crosses being notably sick. These results are consistent with the notion that depletion of GSTT4 alone results in similar phenotypes as observed in our LD-deprived mutants and when combined with *Jabba* mutations causes inviability.

According to published in situ data, *GSS2* mRNAs are maternally provided (BDGP GSS2 probe/CG32495: AT02852). If there is indeed a large pool of these proteins from a maternal source, zygotic knockdown may not be effective. Nevertheless, a zygotically introduced hairpin which targets both genes significantly reduced hatching success in the *Jabba* mutant background (down 23% in *Jabba*$^{L}$ relative to the no-hairpin control) but had no noticeable effect in the wild type (Fig 7F). In the *dPLIN2*$^{-/-}$ backgrounds, there was no statistically significant difference, though mean hatching success rates were lower than wild-type/OrR outcross controls.

Combined, our data indicate that lipid-deprived embryos rely on the glutathione system for efficient hatching success.

## Lipid peroxidation of exogenous lipids is elevated in LD-deprived embryos

One of the functions of the glutathione system is to protect against oxidative damage. Our proteomic and mRNA data indicate an even broader oxidative stress response in our lipid-deprived embryos, with the upregulation of the reducing enzymes Prx6b, Grx1, Prx6C, and SelR (Prx6b and Grx1 Fig 5D, Grx1, Prx6c, and SelR S9 Table) and high significance of GO terms for 'Oxidoreductase activity' (P = 1.63 X 10$^{-9}$). Finally, human orthologs of GSTT4 (hGSTT1 and hGSTT2) are implicated in cancer and epilepsy [39], which may be a result of increased oxidative stress in humans null, or otherwise mutant, for hGSTT1 [39].

The substrate(s) for GSTT4 is unknown and, based on GSTT-family orthologs (GSTT4 annotation on FlyBase), could be hydrogen peroxide, oxidized proteins, or peroxidized lipids. We therefore tested for lipid-peroxidation activity in our LD-deprived mutants. We prepared homogenates from Stage 14–16 embryos, added the lipid-peroxidation sensor BODIPY-C11, and measured the fluorescent properties of the preparation. BODIPY-C11 is a fatty acid analog that exhibits bright red fluorescence in its reduced state. When oxidized, fluorescence emission shifts to green, and thus the ratio of green to red signal provides a measure for the extent to

which it has been oxidized. Compared to the wild type, there was a significant increase in lipid peroxidation in *Jabba*[-/-], *klar*[-/-] and *dPLIN2*[-/-] (Fig 7G). We also performed this assay on newly hatched wild type, *klar*[-/-], and *Jabba*[-/-] L1 larva, and found a significant increase in lipid peroxidation in *Jabba*[-/-] (Fig 7H), and a trend in *klar*[-/-] consistent with additional peroxidation (Fig 7G, p = 0.13). These results indicate that LD-deprived embryos display an oxidative burden that persists even post embryogenesis. Because our mRNAseq and proteomics data indicate that mitochondrial gene products are altered, we looked for evidence of hampered mitochondrial function using the inner membrane potential dye TMRE. We exposed Stage 15 embryo homogenates to the dye TMRE, which fluoresces proportionately to the inner mitochondrial membrane's potential. We found no difference between wild type, *Jabba*[-/-], and *dPLIN2*[-/-] (Fig 7I).

We then tested whether GSTT4 was capable of reconciling the BODIPY-C11 lipid peroxidation. We drove the GSTT4 RNAi in our wild type and *Jabba*[L/L] da-gal4 lines and tested for lipid peroxidation using BODIPY-C11. There was significantly more oxidation of BODIPY-C11 with GSTT4 RNAi in both the wild type relative to wild type with RNAi and *Jabba*[-/-] relative to *Jabba*[L/L] with RNAi (Fig 7G). Thus LD-deprived embryos generate increased levels of BODIPY-C11 lipid peroxidation and zygotic expression of GSTT4 reduces oxidative damage in wild type and *Jabba* mutants.

## LD-deprived mutants have an increased reliance on zygotic ATGL/Bmm

Both the transcriptomic and proteomic datasets are enriched for genes involved in fatty acid and lipid metabolism. For example, in both *Jabba*[-/-] and *dPLIN2*[-/-] mutants, we found an increase in protein levels for the LD protein Stur (a highly conserved LD hydrolase [40]) and FASN1 (fatty acid synthase) (Fig 6B). However, there were also sharp differences, such as for the prominent LD lipase ATGL (also called Brummer in Drosophila), the principal cytosolic triglyceride lipase in higher animals. Even though *ATGL* mRNAs were up for *Jabba*[-/-] and *dPLIN2*[-/-] mutants compared to wild type (Fig 8A), protein levels were only increased in *Jabba*[-/-] embryos, by about ~70% relative to wild type. In *dPLIN2*[-/-] embryos, they were even decreased by ~30% relative to wild type. This discrepancy is surprising, as ATGL is a key enzyme in lipid metabolism and known to be essential for Drosophila embryogenesis [6,10]. Yet it does not appear to be part of a shared response to lipid deprivation.

This discrepancy extends to the functional level. When we used *da-Gal4* lines to drive ATGL RNAi, viability in the *Jabba*[L/L] background dropped from ~80% viability to ~38% (Fig 8B). In contrast, there was no significant change in hatching success for wild type and *dPLIN2*[-/-] embryos, despite the fact that their ATGL proteins are lower. Thus, not only is ATGL protein expression upregulated in *Jabba*[-/-], but embryos are particularly sensitive when its levels are reduced. In *klar*[-/-] embryos, viability upon ATGL knockdown drops from 66% to 14% (Fig 8B), suggesting that zygotic expression of ATGL is uniquely important when LDs are mislocalized to the yolk cell.

Other observations also suggest that aspects of lipid metabolism are different when LDs are mislocalized versus depleted everywhere. PGC1α/Spargel (Srl) is a nuclear transcription factor implicated in a number of lipid signaling pathways. Zygotic SRL RNAi led to a drop in viability in the wild type and in *dPLIN2*[-/-] embryo*s*; however, no such reduction was observed for embryos from *Jabba*[L/L] mothers (Fig 8F).

While all three mutants have reduced LD availability in the periphery, there is one dramatic difference between *dPLIN2* embryos and the LD-allocation mutants: the latter have a normal supply of LDs to support development, just in the wrong tissue. If they were able to access this LD population, it might alleviate their lipid-deprived state.

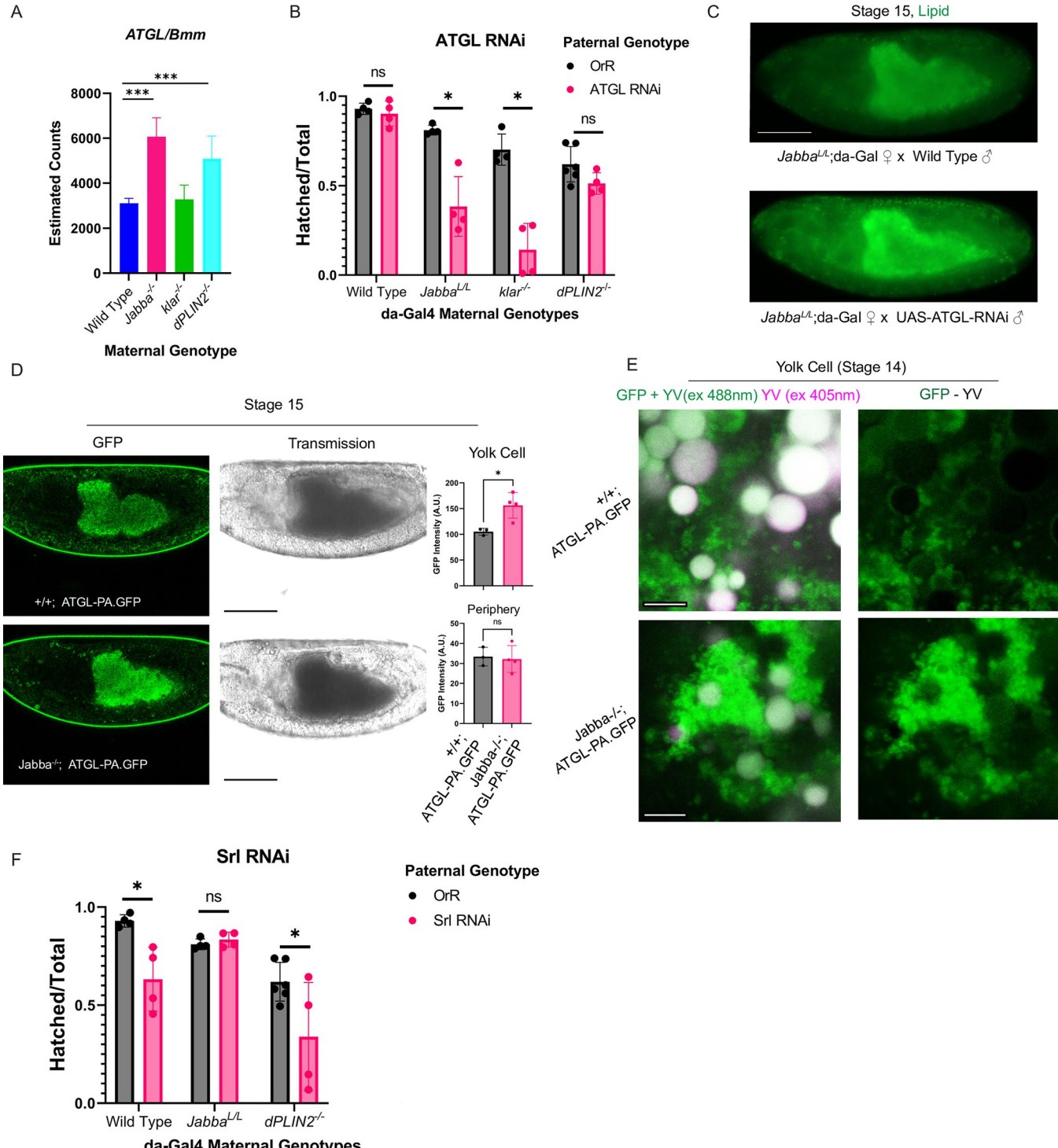

**Fig 8. LD-transport mutant embryos have an increased reliance on zygotic ATGL.** A) Relative mRNA levels for the lipase ATGL/Bmm in wild type and the three LD-deprived embryos. B) Embryonic hatching success for embryos with zygotic RNAi for UAS-ATGL-RNAi driven by *da-Gal4*. N is roughly 100 embryos scored on four different days from at least two sets of crosses per genotype. C) Neutral lipid staining (Nile Red) of Stage 15 embryos from *Jabba^{L/L}*;*da-gal4* mothers crossed to either wild type or UAS-ATGL-RNAi fathers. Note the additional lipid signal in embryos with zygotic ATGL RNAi. Scale bar 100 μm. D) Endogenously tagged ATGL-PA-GFP shows condensed signal in the periphery of *Jabba^{-/-}* embryos relative to wild type, and more intense signal in the yolk cell. Transmission images show the embryos are of the same stage. GFP intensity in the yolk vs the periphery is quantified in the graphs on the right. N is three embryos for wild type and four for *Jabba^{-/-}*. Scale bars 100 μm. E) Endogenously tagged ATGL-PA-GFP shows increased signal in the yolk cell of *Jabba^{-/-}* embryos relative to wild type. As the yolk cell contains autofluorescent, broadly excited yolk vesicles, we first show the unedited GFP channel with a 405nm

channel (peak yolk vesicle excitation) super imposed to show the signal attributable to yolk vesicles in white. The next images have that 405nm channel subtracted from the GFP channel to only show signal from GFP. Scale bars 5 μm. F) Embryonic hatching success for embryos experiencing RNAi for Srl. Wild-type embryos respond but LD-transport mutants do not, showing that our RNAi system's phenotypes are attributable to interactions between the maternal background and RNAi target, and not a general sensitization of our mutants to RNAi. N is roughly 100 embryos scored on four different days from at least two sets of crosses per genotype. B,D,F) Significance was determined with unpaired Student's t-tests in GraphPad Prism, ns is P > 0.05, * is P ≤ 0.05, ** is P ≤ 0.01, *** is P ≤ 0.001. A) significance was determined by DEseq2's p-adjusted value using the 3 replicated per genotype. Error bars represent standard deviation.

To gain further insight into the ATGL discrepancy, we examined the distribution of ATGL across the embryo. We utilized a new ATGL allele in which the endogenous PA isoform is C-terminally tagged with GFP [41] and introduced it into wild-type and *Jabba*$^{-/-}$ backgrounds. In Stage 15, wild-type embryos displayed dispersed green signal in the periphery (Fig 8D), while *Jabba*$^{-/-}$ mutants had more signal in the yolk cell (Fig 8D, quantification), and the peripheral signal was more condensed (Fig 8C). Because the yolk cell contains broadly autofluorescent yolk protein vesicles, we sought to separate the GFP and yolk autofluorescence to unambiguously detect ATGL-GFP. High magnification images of the GFP channel were taken in the yolk cell at Stage 14, when the yolk cell is pressed up against the amnioserosa just beneath the eggshell (Fig 8D). Individual LDs could be made out at this stage (Fig 8E), confirming that we could resolve GFP positive LDs versus yolk vesicles. Then images were taken with excitation at 405nm to capture only yolk vesicle fluorescence, which were then subtracted from the GFP channel (Fig 8E). This analysis showed that *Jabba*$^{-/-}$ embryos indeed contain much more ATGL-PA-GFP in the yolk cell than their wild-type counterparts.

To test whether zygotic ATGL expression in the *Jabba*$^{L/L}$ background is consuming LDs, we performed neutral lipid staining on embryos with and without ATGL RNAi. Embryos with ATGL RNAi had more LD staining (Fig 8C), supporting the idea that *Jabba* embryos upregulate ATGL expression from the zygotic genome to consume their misallocated LDs and survive-LD deprivation.

## Discussion

Here we provide the first evidence that LD allocation amongst embryonic lineages impacts subsequent development. At fertilization, LDs are found homogenously throughout the early Drosophila embryo, but by cellularization they are asymmetrically distributed between cell types: we estimate that 80% of all LDs are present in the peripheral tissue, with the rest allocated to the yolk cell. This uneven inheritance requires an elaborate machinery: actin-driven cytoplasmic streaming, the anti-clustering activity of Jabba to prevent LDs from sticking to glycogen granules, kinesin-1 and cytoplasmic dynein-driven motion of LDs along microtubules, and regulators like Klar, Halo, and Wech that ultimately control the balance of kinesin and dynein activity on LDs [1,10,11,15,16,17,42]. Yet until now the physiological relevance of these complex motions was unknown. We disrupted two distinct mechanisms of allocation using mutants in *Jabba* and *klar*, which results in embryos with normal numbers of LDs sorted into the wrong lineage. These embryos show numerous signs of metabolic distress as well as delays in embryogenesis. Using *dPLIN2* null embryos as an LD-deprived, but appropriately localized, reference, we show that mis-allocation mimics decreased maternal loading of LDs. Our data is the first to demonstrate that early embryonic LD lineage allocation ensures their accessibility and supports later metabolic success, punctual development, and protects redox homeostasis. In particular, we show that where LDs are in the nascent body plan is an important player in their utilization (S1 Graphical Abstract), comparable in importance to receiving a full LD supply from their mother. As animal embryos of diverse species sort LDs very early in embryogenesis (Fig 1), we propose that early embryonic asymmetric LD allocation is a conserved means of sorting LDs into those embryonic lineages that require them for timely development.

## Embryos can survive despite a substandard nutrient supply

Embryos from oviparous animals, including Drosophila, are a closed system unable to take up nutrients from the environment. All the nutrients required for embryogenesis must therefore be provided by the mother to the oocyte. Our measurements suggest that during normal Drosophila development most of the TAG reserves inherited from the mother are consumed by the end of embryogenesis (Fig 3A–3C), implying that the mother endows the embryo with just enough lipid reserves to support development until the larva can feed. It is therefore surprising that the majority of *Jabba*[-/-] and *klar*[-/-] embryos hatch successfully, even though they cannot access all their neutral lipid stores and retain them into larval stages. Similarly, even though *dPLIN2*[-/-] embryos have only ~66% of the wild-type triglyceride supply, ~60% still successfully complete embryogenesis (and most of the rest reach Stage 17).

Intriguingly, a parallel situation is observed with the maternal supply of glycogen. Glycogen reserves are largely consumed during Drosophila embryogenesis [5,28], yet embryos laid without any glycogen (in mothers mutant for GlyS, glycogen synthase) hatch at near wild-type levels [43]. Similarly, embryos that have glycogen reserves but cannot access them (because of the absence of GlyP, glycogen phosphorylase) hatch successfully [43]. Somehow, Drosophila embryos are able to hatch even if they lack a substantial portion of their maternally provided energy stores. We hypothesize that they achieve metabolic flexibility by switching between energy substrates if needed. Indeed, embryos lacking glycogen show reduced levels of acyl carnitines [43], suggesting increased lipid breakdown to compensate for the lack of carbohydrates. In turn, our lipid-deprived embryos upregulate carbohydrate-related genes, including the trehalose transporter Tret1-1, and Trehalase (Figs 7A–7C and 6B).

Metabolic flexibility, including flexibility in substrate utilization, is a trait previously described in mature organisms in mammals and flies [44,45,46,47,48]. Here, the animal can utilize multiple biological molecules to accomplish the same goal. For example, both glucose and lipid-derived products (like ketone bodies or fatty acids) can be used in energy production. Substrate switching is integral to surviving some of the most common human diseases, including diabetes [48], neurodegenerative diseases [49], and heart disease [50]; individuals with a metabolic disease have dramatic alterations, for example, utilizing lipid-derivatives in place of glucose in individuals with diabetes [48].

We propose that LD- or glycogen deprived embryos may similarly be able to rewire their metabolism to more heavily rely on other substrates. A detailed test of this hypothesis will require interrogating carbohydrate levels in a lipid deprived background, investigating the contribution of amino acid consumption (glutamate, aspartate, and glycine are thought to be consumed [6,51]), and measuring oxygen consumption during embryogenesis to determine if hatching can be achieved with less overall respiration or if deprived embryos need to 'burn the furniture' to hatch. While null mutants in *Jabba*, *klar*, *dPLIN2*, *GlyP*, and *GlyS* can hatch successfully, our attempts to generate several of the double mutant combinations failed, consistent with the idea of compensation between energy sources.

Even though LD-deprived embryos are able to hatch, they are compromised. Starvation is often associated with changes in oxidative damage [52,53,54], and multiple lines of evidence suggest that LD-deprived embryos are challenged in maintaining their redox balance. First, using BODIPY-C11, we find all three LD-deprivation mutants have a significant increase in lipid peroxidation. Second, genome-wide analysis indicates that the mutants upregulate many transcripts and proteins involved in redox homeostasis, as if they need to combat redox stress. Finally, *Jabba*[-/-] *mutants*, but not wild type, have a significant drop in viability when glutathione synthase or glutathione transferase T4 are zygotically knocked down.

### An active, beneficial response to lipid starvation

We find that embryos respond to lipid deprivation at the transcriptional and proteome level. We detect robust signals in oxidative stress, sugar metabolism, nucleotide metabolism, amino acid metabolism, iron metabolism, sulfur containing proteins, ubiquitin conjugation systems, ER and mitochondrial genes at the mRNA level and peptide level. In principle, the observed changes might be a consequence of a breakdown in gene regulation for embryos struggling to survive. However, we favor the idea that this response is largely adaptive, benefitting the embryo by allowing it to overcome the metabolic challenge. The upregulation of several key players in trehalose metabolism is certainly consistent with the idea of compensating for the lack of lipids by upregulating carbohydrate utilization. And for two upregulated enzymes, ATGL and GSTT4, we show that they benefit the lipid-deprived embryo, as RNAi for both genes diminishes hatching. Importantly, this sensitivity of lipid-deprived embryos to the loss of certain factors is specific and does not indicate that they are simply impaired across the board; while knockdown of the lipid responsive transcription factor Srl/PGC1alpha reduced the viability of wild type and *dPLIN2*<sup>-/-</sup> embryos, it was not detrimental to *Jabba*<sup>-/-</sup> embryos. We conclude that embryos can mount an adaptive response to lipid deprivation that ensures efficient hatching. To fully characterize the breadth of this adaptation in the future, it will be important to develop additional measures for the physiological state of these embryos and test which parameters are compromised when candidate response genes are knocked down.

The oxidative stress we detected in our LD-deprived embryos suggests an intermediate source of oxidative damage that is activated when LDs are absent. Likely, this is the mitochondria which are the site of breakdown of fatty acids released from LDs [1], a source of cellular ROS [55], and implicated in both our mRNAseq and proteomics data as being affected in our mutants. Specifically changes in mitochondrial proteins handling metals such iron (ferritin light and heavy chain, IRP 1A, MagR, etc.) and copper (GRX1) suggest LD-deprivation causes insults in key mitochondrial enzyme complexes. We suspect that in our LD deprived mutants fatty acid import into the mitochondria is reduced, leading to the downstream effects on the mitochondrial proteome. However, our attempts to perturb the mitochondrial fatty acid importer CPT1/Whd through zygotic knock down lead to no detectable effects, possibly because CPT1 is abundantly supplied maternally [56]. Double mutants for the *whd*<sup>1</sup> and *klar*<sup>YG3</sup> alleles were inviable, supporting a link between LD allocation and mitochondrial fatty acid consumption.

The existence of this elaborate response suggests that in nature newly laid embryos are not always laid with a full complement of nutrients and thus need to be able to adapt to survive. The nutritional state of the mother certainly can have a strong effect on her overall fecundity and the number of eggs laid [57,58]. Although there are homeostatic mechanisms that result in similar endowment of oocytes with triglycerides despite varying maternal nutrition [59], it is not known how well they can keep nutrient levels stable across conditions, especially in natural environments where nutrient availability likely fluctuates widely.

### Hatching delay may be a general response to nutrient deprivation

Lipid-deprived embryos also exhibit delayed hatching. Most of the delay occurs after Stage 15, when lipid deprivation has already resulted in a dramatic response at the transcriptomic and proteomic levels. Thus, the delay is not what triggers the adaptive response of the embryo; rather, embryos are able to sense their nutrient-deprived state by an as-yet uncharacterized mechanism. Intriguingly, a hatching delay of similar magnitude is observed in embryos lacking the majority of their glycogen supply [43]. As embryos contain roughly 10x the calorie supply in LD-stored TAG as found in glycogen [5], these glycogen mutants (*GlyP* and *GlyS*) are in

similar state of gross calorie deprivation to our *Jabba*[-/-] mutants which hatch with ~15% of their TAG unconsumed and another ~60% consumed in the wrong tissue, and *dPLIN2*[-/-] mutants which receive ~30% less TAG from their mother [26]. A developmental delay may therefore be a general response embryos employ when their calorie supply is diminished. It will be interesting to determine whether glycogen-deprived embryos also respond with drastic changes to their transcriptome and proteome and to what extent this response differs from that observed upon lipid deprivation. Such a comparison may shed light both on how suboptimal nutrition is sensed and how it leads to a developmental delay.

Curiously, *dPLIN2* mutants are more severely delayed than *Jabba*[-/-] and *klar*[-/-] mutants, even though LD starvation of the periphery appears worse in the latter genotypes. We estimate based on quantification of LD abundance in TEMs immediately post cellularization, that for the latter genotypes ~70% of LDs that would normally be in the periphery are shifted to the yolk cell. In contrast, *dPLIN2*[-/-] embryos have a global ~30% reduction in LDs, in both the periphery and yolk cell [26]. Why, then, are *Jabba* and *klar* mutants not more severely affected if their peripheral cells are more LD deprived? Our analysis suggests that these mutants partially overcome this peripheral deficit by consuming a portion of the inappropriately localized LDs via upregulation of the lipase ATGL. Consistent with this idea, TAG measurements in *Jabba*[-/-] and *klar*[-/-] show that only ~1/4[th] of excess LDs deposited are not consumed. And ATGL knockdown in *Jabba*[-/-] and *klar*[-/-] embryos interferes with yolk LD consumption.

ATGL is essential for Drosophila embryogenesis [6] and is the major cytoplasmic enzyme which liberates TAG for consumption. We find that *Jabba*[-/-] and *klar*[-/-] embryos, but not *dPLIN2*[-/-], require additional transcription of ATGL for survival, relative to wild type (Fig 8B). Notably, wild type does not require zygotic expression of ATGL at all, if provided maternally [6]. *Jabba*[-/-] and *klar*[-/-] embryos may, then, employ boosted zygotic levels of ATGL to increase the lipolytic capabilities of the yolk cell. Thus, flexible utilization of ATGL in *Jabba*[-/-] and *klar*[-/-] mutants appears to be able to compensate for a more severe LD-deprivation in the periphery.

In summary, our analysis provides the first evidence that the nutrient layout of the embryo plays a role in the metabolic success of embryogenesis (see S1 Graphical Abstract). Having LDs in the wrong tissue is akin to receiving a reduced maternal supply. Further, we provide evidence that LD deprivation causes oxidative stress and delays embryogenesis. Finally, we show that embryos are capable of overcoming LD deprivation through flexible utilization of ATGL and glutathione metabolism. In the absence of asymmetric allocation, mothers could presumably achieve an adequate LD supply for the developing epithelium by increasing the overall number of LDs loaded into the egg. However, this strategy would result in a yolk cell with superfluous, unused LDs and thus wasted resources. Asymmetric inheritance of LDs allows mothers to optimize their LD investment per embryo without compromising successful embryogenesis.

## Methods

### Origin of fly strains

Oregon R was used as the wild-type strain. *Jabba*[DL], referred to as *Jabba*[-/-], was generated previously in the laboratory and is a strong loss-of-function allele with no Jabba protein detected in early embryos [21]. *Jabba*[Low] is a hypomorphic allele with a strong reduction in protein production [24]. *klar*[-/-] refers to the *klar*[YG3] allele generated previously in our lab; although this loss-of-function allele does not eliminate all Klar isoforms, it abolished the klar β isoform that regulates LD transport in early embryos [60]. The strain referred to as ATGL-RNAi was obtained from the Vienna Drosophila Resource center (VDRC ID 37880). The strains referred

to as GSTT4 RNAi (BL# 36717), GSS RNAi (BL# 55150), SRL RNAi (BL# 33915), GSTT4$^{-/-}$ (BL#77777), and *whd*$^1$ (BL#441) were obtained from the Bloomington Drosophila Stock Center (BDSC). LpR1$^{-/-}$ and Lpr1$^-$,LpR2$^-$/Tm3 were generated by Joaquim Culi's group [29]. The *dPLIN2*$^{-/-}$ allele is the protein null allele *LSD-2*$^{KG}$ previously described [27] The *da-Gal4* lines were generated from Bloomington stock BL#55850 which has a 3$^{rd}$ chromosome insertion, except for *klar* which used a 2$^{nd}$ chromosome insertion of the same construct generously provided by Elizabeth Knust (FlyBase ID FBti0162419). ATGL-PA-GFP flies [41] were generously provided by Xun Huang.

## Publicly available resources

FlyBase [61] was utilized for numerous genetic and curation resources. Drosophila embryo mRNA expression data used are from the Berkley Drosophila Genome Project (BDGP) [62,63,64]. GOrilla [65,66] was utilized for GO analysis of the proteomics data set. FlyEnrichr [67,68] was utilized for GO analysis of the mRNA data set.

## Microscopy

Laser scanning confocal microscopy was performed on a Leica SP5 equipped with HyD detectors, using either a 40x objective to show most of the embryo, or a 63x objective for subregions. Spinning disc confocal microscopy was performed on an Andor Dragonfly in the University of Rochester High Content Imaging Core and was used to capture the mud snail embryo images with a 40x objective. Epifluorescence imaging was performed on a Nikon Eclipse E600 using a 20x objective, Spot Insight imaging software, and a Diagnostics Instruments' 14.2 Color Mosaic camera. All images were assembled in Adobe Illustrator.

S1–S3 Videos were captured at 1 frame per 2 seconds, 512x512 pixels. Video processing was done using FIJI (NIH).

Sample sizes were as follows. For live imaging, at least three embryos were imaged per genotype per experiment. For fixed samples, the stainings were performed at least twice, with multiple embryos imaged per staining. For TEM analysis, the core facility was given ten appropriately staged embryos per genotype per experiment, and then chose which were imaged based on staining success.

Exclusion criteria for imaging embryos were predetermined. Embryos not of the stage of interest, determined to have expired during preparation or image acquisition, or which were imaged in the incorrect orientation/focal depth were excluded.

## Periodic acid Schiff (PAS) and LD staining

Embryos were collected on apple juice plates for the desired time range and dechorionated with 50% bleach and fixed for 20 minutes using a 1:1 mixture of heptane and 4% formaldehyde in phosphate-buffered saline (PBS). To detect GGs, embryos were devitellinized using heptane/methanol and subsequently washed three times in 1xPBS/0.1% Triton X-100. Embryos were incubated first in 0.1M phosphatidic acid (pH 6) for 1hr and then in 0.15% periodic acid in dH$_2$O for 15min. After one wash with dH$_2$O, embryos were incubated in Schiff's reagent (Sigma-Aldrich) until the embryos went from uncolored, to pink, to red (~2 minutes). To stop the reaction, it was quenched with 5.6% sodium borate/0.25 normal HCl stop solution for at least 2 minutes with agitation.

After replacing half the volume of the stop solution with an equal volume of 1×PBS/0.1% Triton X-100 to reintroduce detergent, the sample was shaken vigorously to free embryos stuck to the container or each other. For subsequent imaging, embryos were mounted in either Aqua-Poly/Mount from Polysciences or a glycerol solution (90% glycerol, 10% PBS) since

mounts with 'antifade' or $O_2$ scavenging additives should be avoided for this approach. Fluorescent PAS signal was then detected using an Alexa633 channel on a Leica Sp5.

To detect LDs by staining, the methanol step for devitellinization needs to be omitted as it extracts neutral lipids. Instead, after fixing with heptane/formaldehyde, the embryos were washed extensively in a wire mesh basket with 1×PBS/0.1% Triton X-100 to remove residual heptane, then transferred to a 1.7mL microcentrifuge tube. They were then washed 2x with 1×PBS/0.1% Triton X-100. For costaining with PAS, embryos were incubated in 0.1M phosphatidic acid (pH 6) plus 1μL of 1mg/mL BODIPY 493/503 (Invitrogen) for 1 hr; then the same steps as for PAS above were followed (starting with periodic acid incubation). As red lipid dyes overlap Schiff's reagent's spectra, we avoided them for costaining with PAS.

For LD staining without PAS costaining, the 1×PBS/0.1% Triton X-100 wash was removed and replaced with 1×PBS/0.5% Triton X-100/10% BSA/0.02% sodium azide; embryos were incubated for 1 hr. The solution was replaced with 1 ml of fresh 1×PBS/0.5% Triton X-100/ 10% BSA/0.02% sodium azide and then either 1μL of 1mg/mL BODIPY 493/503 in DMSO, 1 μL LipidSpot 610 (1000x) (Biotium) in DMSO, or 10μL of 200mg/mL Nile Red (Sigma Aldrich) in acetone was added. After 20 min of staining, embryos were washed and mounted as in the previous paragraph.

## Live imaging

For live imaging involving dye injections, a previously published procedure was followed [20]. In short, embryos were collected on apple juice plates for the desired time, hand-dechorionated, transferred to a coverslip with heptane glue, desiccated, and placed in Halocarbon oil 700. Embryos were then injected with DMSO containing BODIPY 493/503 (1mg/mL), SIR Tubulin or SPY Tubulin and imaged on a Leica Sp5 confocal microscope.

## Immunofluorescence

Embryos of the desired stage were collected, dechorionated with 50% bleach, and fixed for 20 min using a 1:1 mixture of heptane and 4% formaldehyde in phosphate-buffered saline (PBS). Embryos were devitellinized using heptane/methanol, followed by four methanol washes. Embryos were then washed in 1×PBS/0.1% Triton X-100 several times to remove residual methanol. Embryos were blocked with ovary block (1×PBS/0.5% Triton X-100/10% BSA/ 0.02% sodium azide) for at least an hour. Anti-Tret1-1-PA antibody was a generous gift from Dr. Stefanie Schirmeier [32]. The anti-Tret1-1-PA antibody was diluted 1:1000 in ovary block, 1mL of which was added to the embryos which were then incubated overnight at 4˚C. Embryos were then washed 5x with 1×PBS/0.1% Triton X-100 over the course of 30min. Embryos were then stained with secondary antibodies (Invitrogen goat anti-rabbit or goat anti-mouse tagged with either Alexa633 or Alexa488) at 1:1000 in ovary block overnight at 4˚C. Embryos were then washed 5x with 1×PBS/0.1% Triton X-100 over the course of 30min. Embryos were mounted in Vectashield with DAPI (Vectorlabs) and imaged using an Andor Dragonfly.

## Neutral lipid TLC and enzymatic TAG assays

300 embryos of the desired age were collected and placed into 2mL microcentrifuge tubes on ice. Then the samples were flash frozen in liquid nitrogen. They were then thawed, the buffer removed, and 200μL of TSS (1×PBS/0.1% Triton X-100) was added. The embryos were then homogenized on ice using a motorized pestle and mortar and placed at 70˚C for 3min to denature lipases. The samples were spun at max speed for 3min. 10μL of heat-treated homogenate was saved at -80˚ C for a Bradford assay.

For TLC analysis, the homogenate was transferred to a 0.5mL tube; then an equal volume of chloroform:methanol (2:1) was added. The tube was sealed with Parafilm to prevent evaporation, then placed on a rotator overnight at 4˚C. The tubes were spun at max speed for 5min. Then the clear, bottom fraction (the chloroform: methanol) was collected into a new tube while avoiding the precipitated protein laying at the interphase. The same volume was collected for each sample if pairwise comparisons were made. The samples were then dried under a vacuum. The samples were resuspended in 50 µL chloroform: methanol (2:1) and analyzed immediately. The TLC tank (or large beaker) was prepared by placing the solvent inside and sealing the tank. The tank was lined with filter paper to increase the evaporative surface area and improve resolution. Neutral lipid solvent: Petroleum Ether/Diethyl Ether/acetic acid (32:8:0.8) (e.g., 160 µL PE + 40µL DEE + 4µL AA). The tank was sealed for at least 30min to allow vapor to fill the tank. The TLC plate was cut to desired size (TLC silica gel 60 plates (EMD Millipore 1055530001)). TLC plates were dehydrated in 100˚C oven for 30min. Lanes were marked on the plate with pencil and ruler (leaving about 3 cm between each lane and the edges of the plate) and at least 3cm from the bottom of the plate was allowed. Care was taken to avoid submerging samples into the TLC solvent and to make sure the plate was level. Samples were spotted (1 µL increments) into each lane to a desired volume and the solvent was allowed to evaporate between spotting increments. The TLC plate was placed into the TLC tank solvent as quickly as possible to prevent escape of solvent vapor. Care was taken that the spots were not submerged into the solvent. The solvent was then allowed to run to almost the top of the plate. The plate was removed from the TLC tank and the solvent was given time to evaporate from the plate. The TLC tank can be resealed, and the solvent reused. The dried TLC plate was then submerged in charring solution (50% ethanol, 3.2% H2SO4, 0.5% MgCl2). The plate was charred in a 120˚C oven for 30min and imaged. Band quantification was done in FIJI (NIH). The TAG band value was divided by sterol band value to control for plate-to-plate variations in charring. This TAG/STEROL ratio was then standardized against protein readings from a Bradford assay performed on the crude homogenate to control for sample input.

For enzymatic assays, 20µL of heat-treated homogenate was transferred into a tube containing either 20µL of Triglyceride Reagent (Sigma) or 20µL of PBS. Both PBS and TAG Reagent were required for each sample; the PBS controls for free glycerol. The samples were run in triplicate. They were then incubated for 30min at 37˚C. The samples were then centrifuged at max speed for 3min, then 30µl per sample were transferred to a 96-well plate with 100µL glycerol free reagent (Sigma) per well and incubated for 5min at 37˚C. The absorbance was measured using a plate reader set to 540nm. The TAG concentration was determined for each sample by plotting the absorbance on a standard curve based on a glycerol standard (Sigma), then subtracting the PBS sample from the Triglyceride reagent sample. The readings were then controlled for input using protein levels from a Bradford assay.

## BODIPY C11 lipid peroxidation assay and TMRE assay

Embryos or newly hatched L1 larva (which were less than 2hrs post hatching) were treated in 50% bleach for two minutes, then collected in 50µL of TSS (0.4%w/v NaCl, 0.03%v/v Triton X-100). 10 individuals were collected per replicate. The BODIPY C11 (Thermo) solution was made by adding 1µL of 10µg BODIPY C11/µL DMSO to 2mL of Grace's Insect Medium (Sigma-Aldrich). 100 µL of Grace's Medium + C11 were then added to each replicate. These samples were homogenized for 20s using a motorized pestle. Samples were briefly spun down then allowed to incubate at 21˚C in the dark for 30min. Samples were then measured on a Molecular Devices' SpectraMax M2e, set to capture two fluorescent readings: red excitation

561nm/ auto cutoff 590/ emission 591 and green excitation 485nm/ auto cutoff 495/ read 510nm. Data is represented as the green reading divided by the red reading.

TMRE assays were performed with the above protocol with 1μL of 10μM TMRE (Biotium) in DMSO replacing the BODIPY C11. Samples were imaged without and incubation step on a Molecular Devices' SpectraMax M2e set to capture one reading at excitation 549nm/ cutoff 570nm/ emission 575nm.

### GSH-GLO assay

GSH-GLO kits were obtained from Promega. The 'Tissue extract' protocol was followed. Embryos were treated in 50% bleach for two minutes, then collected in 50μL of TSS (0.4%w/v NaCl, 0.03%v/v Triton X-100). Six individuals were collected per replicate. The embryos were homogenized in 200μL of PBS, 20mM EDTA. 50 μL of the homogenate was then added to 3 wells on a 96 well plate to allow for triplicate measurements. 50 μL of the 2x GST reaction buffer was added to each well, then incubated for the recommended amount of time (30 minutes) in the dark. The Luciferin Detection Reagent was then added and allowed to incubate for 30 minutes. Luminosity was detected using the Luminosity mode on a Molecular Devices' SpectraMax M2e.

To determine the internal GSH concentration in embryos, the concentration from the standard curve (provided with kit) was used to determine the concentration within the well (reported in μM/Rxn in the Raw graph). The wells contained 100 μL of assayed homogenate, concentrations were then converted to the internal volumes of the 1.5 embryos worth of homogenate provided per well, or 13.95nL for wild type/OrR, 13.14 nL for *Jabba$^{-/-}$*, 13.83nL for *klar$^{-/-}$*, or 13.05nL of *dPLIN2$^{-/-}$* embryos. Volumes were determined using a calibrated microscope.

### Transmission electron microscopy

Embryos were collected from 7- to 14-day-old flies, dechorionated in 3% sodium hypochlorite, and washed extensively with distilled water. Embryos were fixed in 4%paraformaldehyde/2% glutaraldehyde/PBS with an equal volume of heptane added. The vials were shaken, then left on an agitator for 20 minutes. After fixation, embryos were washed extensively with 1×PBS/ 0.1% Triton X-100, then transferred onto a piece of double-sided tape, adhered, then submerged with 1×PBS/0.1% Triton X-100. The embryos were then gently hand rolled using fine forceps until the vitelline membrane was removed. Embryos were transferred to a small glass vial. The embryos were then fixed a second time with 4%paraformaldehyde/2%glutaraldehyde/PBS, excluding the heptane, for 30 minutes. Embryos were then washed three times with 0.2 M sucrose in 0.1M cacodylate buffer. They were washed an additional 3 times in 0.1 M sodium cacodylate before post fixation in 1% osmium tetroxide for 2 hours followed by uranyl acetate enhancement in 0.5% uranyl acetate overnight at 4˚C. Specimen were washed and then dehydrated in a graded ethanol series, transitioned to propylene oxide, and embedded in an Epon/Araldite resin. Thin sections were stained with 0.3% lead citrate and imaged on a Hitachi 7650 transmission electron microscope using an 11 MP Gatan Erlanshen CCD camera. TEM work was conducted at the Electron and Cryo Microscopy Resource in the Center for Advanced Research Technologies at the University of Rochester.

### Hatching assays

Cages of the desired genotype, either isogenic or reciprocal crosses, were set up 2–3 days before collections began to allow the flies to acclimate. The crosses and their relevant controls were always done simultaneously, with male and female siblings being used for the reciprocal

crosses. A pre-collection with yeasted apple juice plates was performed for 1hr, to allow flies to lay fertilized embryos that may have been held. Embryos were then collected for 60-90minutes, then staged under Halocarbon Oil 27 (Sigma), and the experimental and control embryos were transferred with tweezers to the same plate. The plates were aged for either 14hrs at 25°C or 18hrs at 21°C. Hatching was then recorded using a Moticam 2000 camera, capturing 1 image per minute, until all the hatching events had occurred (usually 5hrs). Embryos were then assigned a time post laying and compared between genotypes.

## Zygotic RNAi setups

*Da-Gal4* (3$^{rd}$ chromosome insertion) was crossed into an OrR, *Jabba$^{Low}$*, or *dPLIN2$^{-/-}$* background. Da-Gal4 (2$^{nd}$ chromosome insertion) was crossed into a *klar$^{YG3}$* background. These four genotypes were then used as the maternal background, providing both the lipid droplet phenotype and Gal4. The UAS-sequence (e.g., UAS-ATGL RNAi) was then provided from the father.

## Viability assays

Cages of the desired genotype, either isogenic or reciprocal crosses, were then set up 2–3 days before collections began to allow the flies to acclimate. 10 females and 10 males were the target, though occasional death occurred. Embryos were collected at 25°C for 24hrs, then the plate was removed, then aged at 25°C for 24hrs. The plates were then scored, with the target of scoring 100 embryos per plate. The area scored was defined prior to looking under the dissecting scope to minimize biasing the scored regions. One cross/cage provided two plates, and each cross was performed at least twice.

## RNA sequencing

Virgin females from OrR, *Jabba$^{-/-}$*, *klar$^{-/-}$*, and *dPLIN2$^{-/-}$* were collected and crossed to OrR males. The flies were placed into cages and allowed to acclimate for 2 days. Embryos were then collected and aged at 25°C until they had reached Stage 15. Three Stage 15 embryos (3-tiered midgut Fig 4 Stage 15 and described in [22]) were collected using Halocarbon Oil 27 (Sigma) per genotype, with 3 replicates per genotype. Total RNA was extracted with TRIZOL and chloroform solution. The RNA was precipitated with isopropanol, washed with 70% ethanol, and resuspended in sterile water. The 12 samples were Illumina sequenced by Genewiz. The raw RNA seq data was checked for the read quality and trimmed of the Illumina adapters using FastQC and Trimmomatic respectively. The data was analyzed for the RNA sequencing reads using 2 pipelines independently. For the first pipeline, the data was put through Kallisto's index function to index the genome (6$^{th}$ reference version) downloaded from FlyBase (/genomes/Drosophila_melanogaster/dmel_r6.42_FB2021_05/fasta/dmel-all-gene-r6.42. fasta). They were then mapped to the indexed genome also using Kallisto with the bootstrap value of 100. The read abundance between genetic backgrounds was subjected to DESeq2 for differential expression analysis. For the second pipeline, the data was indexed using the above genome using the bowtie2-build function. Next, the cleaned reads were mapped to the indexed genome also using Bowtie2. Then, they were converted from the Bowtie2 output SAM files to BAM files using SAMtools. The BAM files were sorted and indexed by chromosomal position using the SAMtools sort and SAMtools index functions. Htseq was used to count the number of read abundancy in the sort and indexed mapped BAM files. The gtf file used for htseq was dmel-all-r6.42.gtf. The reads were then mapped using Kallisto, to the Drosophila genome. Differential gene expression comparisons were then done between genetic backgrounds using Deseq2, as in the first pipeline.

## Mass spec analysis

Protein Extraction: Ground fly embryos (30 per replicate) were resuspended in a denaturing lysis buffer; 50 mM triethylammonium bicarbonate (TEAB) (Fischer Scientific) and 5% sodium dodecyl sulfate (SDS). Homogenization and genomic DNA shredding were achieved by high-energy sonication (Qsonica), amplitude 30 and 10s on/60s off, on ice. Samples were clarified of cell debris by centrifugation at 16,000g for 10 min and protein concentration was quantified by the bicinchoninic acid (BCA) assay (Thermo).

Trypsinization: Samples were diluted to 0.25 mg/mL in 5% SDS, 100 mM TEAB, and 10 µg of protein from each sample was reduced with dithiothreitol to 2 mM, followed by incubation at 55˚C for 60 minutes. Iodoacetamide was added to 10 mM and incubated in the dark at room temperature for 30 minutes to alkylate the proteins. Phosphoric acid was added to 1.2%, followed by six volumes of 90% methanol, 100 mM TEAB. The resulting solution was added to S-Trap micros (Protifi) and centrifuged at 4,000 x g for 1 minute. The S-Traps containing trapped protein were washed twice by centrifuging through 90% methanol, 100 mM TEAB. 1 µg of trypsin was brought up in 20 µL of 100 mM TEAB and added to the S-Trap, followed by an additional 20 µL of TEAB to ensure the sample did not dry out. The cap to the S-Trap was loosely screwed on but not tightened to ensure the solution was not pushed out of the S-Trap during digestion. Samples were placed in a humidity chamber at 37˚C overnight. The next morning, the S-Trap was centrifuged at 4,000 x g for 1 minute to collect the digested peptides. Sequential additions of 0.1% TFA in acetonitrile and 0.1% TFA in 50% acetonitrile were added to the S-trap, centrifuged, and pooled. Samples were frozen and dried down in a Speed Vac (Labconco) and resuspended in 50 µL of 0.1% TFA.

Data Collection: Peptides were injected onto a homemade 30 cm C18 column with 1.8 um beads (Sepax), with an Easy nLC-1200 HPLC (Thermo Fisher), connected to a Fusion Lumos Tribrid mass spectrometer (Thermo Fisher). Solvent A was 0.1% formic acid in water, while solvent B was 0.1% formic acid in 80% acetonitrile. Ions were introduced to the mass spectrometer using a Nanospray Flex source operating at 2 kV. The gradient began at 3% B and held for 2 minutes, increased to 10% B over 7 minutes, increased to 38% B over 64 minutes, then ramped up to 90% B in 5 minutes and was held for 3 minutes, before returning to starting conditions in 2 minutes and re-equilibrating for 7 minutes, for a total run time of 90 minutes. The Fusion Lumos was operated in data-independent mode. The cycle time was set to 3 seconds. Monoisotopic Precursor Selection (MIPS) was set to 'Peptide'. The full MS1 scan was done over a range of 390–1010 m/z, with a resolution of 60,000 at m/z of 200, an AGC target of 4e5, and a maximum injection time of 50 ms. Precursor ions with a charge state between 2–3 were picked for fragmentation using a staggered windowing scheme of 14m/z with 7 m/z overlaps. Between MS1 scans, a total of 23 MS2 scans over a range of 200–2000 m/z were performed. Precursor ions were fragmented by higher energy C-trap dissociation (HCD) using a collision energy of 33%. MS2 scans were collected in the orbitrap with a resolution of 15,000, an AGC target of 4e5 and a maximum injection time of 23 ms. The number of DIA segments was set to achieve 5–6 data points per peak.

Data Analysis: The raw data were processed with DIA-NN version 1.8.1 [69]. For all experiments, data analysis was carried out using library-free analysis more in DIA-NN using the D. melanogaster UniProt 'one protein sequence per gene' database (UP000000803_7227, downloaded 4/27/2021) was used to annotate the library with 'deep learning-based spectra and RT prediction' enabled. MBR was enabled, which instructs DIA-NN to follow the two-step workflow as described previously. Briefly, DIA-NN first generates a spectral library from the DIA data using all identifications in the specified raw files. The library generation is performed using both global (experiment-wide) and run specific precursor FDR filters of 1%. This library

is then used as a spectral library in a second search. For precursor ion generation, the maximum number of missed cleavages was set to 1, maximum number of variable modifications to 1 for Ox(M), peptide length range to 7–30, precursor charge range to 2–3, precursor m/z range to 400–1000, and fragment m/z range to 200–2000. The quantification was set to 'Robust LC (high precision)' mode with RT-dependent median-based cross-run normalization enabled, protein inferences set to 'Genes', and 'Heuristic protein inference' turned off. MS2 and MS1 mass accuracies were set to 21 and 7 ppm, respectively, and scan window size set to 7. Precursors were subsequently filtered at library precursor q-value (1%), library protein group q-value (1%), and posterior error probability (50%) and protein quantification carried out using the MaxLFQ algorithm as implemented in the DIA-NN R package [70]. Proteins quantified with only 1 precursor and proteins with 2 or more missing values per group were removed and statistical analysis was performed in Perseus 1.6.15.0.

## Statistics

Student's t-tests were chosen for comparisons, as the analyses were designed to address whether two populations of related numbers were statistically different (e.g., hatching for embryos from *Jabba^{L/L}; da-gal4* mothers with or without ATGL RNAi). The t-tests were unpaired and two tailed. Tests were performed on embryos or embryo derivatives (e.g., the homogenates used in the GSH-GLO assay) which were treated identically.

## Supporting information

**S1 Table. RNAseq Differentially expressed in all 3 mutants list.** Genes differentially expressed in all 3 lipid droplet mutant embryos (*Jabba^{-/-}*, *klar^{-/-} and dPlin2^{-/-}*). The columns from left to right show the gene name, the log$_2$ fold change in the Jabba_vs_wild type comparison, the p adjusted value in the Jabba_vs_wild type comparison, the log$_2$ fold change in the Klar_vs_wild type comparison, the p adjusted value in the Klar_vs_wild type comparison, the log$_2$ fold change in the dPLIN2_vs_wild type comparison, the p adjusted value in the dPLIN2_vs_wild type comparison.
(XLSX)

**S2 Table. RNA seq Jabba vs Wt all genes DE.** Genes differentially expressed in the *Jabba^{-/-}* vs wild type comparison. The columns from left to right show the Flybase gene name, the mean of normalized counts for all samples, the log$_2$ fold change, the standard error for the log$_2$ fold change, the Wald statistic, p-value, p-value adjusted for multiple testing, and current gene name.
(XLSX)

**S3 Table. RNA seq klar vs Wt all genes DE.** Genes differentially expressed in the *klar^{-/-}* vs wild type comparison. The columns from left to right show the Flybase gene name, the mean of normalized counts for all samples, the log$_2$ fold change, the standard error for the log$_2$ fold change, the Wald statistic, p-value, p-value adjusted for multiple testing, and current gene name.
(XLSX)

**S4 Table. RNA seq LSD2(dPLIN2) vs Wt all genes DE.** Genes differentially expressed in the *dPLIN2^{-/-}* vs wild type comparison. The columns from left to right show the Flybase gene name, the mean of normalized counts for all samples, the log$_2$ fold change, the standard error for the log$_2$ fold change, the Wald statistic, p-value, p-value adjusted for multiple testing, and current gene name.
(XLSX)

**S5 Table. File RNA seq Jabba vs Wt all genes count.** The counts for the genes detected in *Jabba*[-/-] and wild type. The columns from left to right show the Flybase gene name, the estimated counts for *Jabba* replicate 1, the estimated counts for *Jabba* replicate 2, the estimated counts for *Jabba* replicate 3, the estimated counts for wild type replicate 1, the estimated counts for wild type replicate 2, the estimated counts for wild type replicate 3.
(XLSX)

**S6 Table. RNA seq klar vs Wt all genes count.** The counts for the genes detected in *klar*[-/-] and wild type. The columns from left to right show the Flybase gene name, the estimated counts for *klar* replicate 1, the estimated counts for *klar* replicate 2, the estimated counts for *klar* replicate 3, the estimated counts for wild type replicate 1, the estimated counts for wild type replicate 2, the estimated counts for wild type replicate 3.
(XLSX)

**S7 Table. RNA seq LSD2(dPLIN2) vs Wt all genes count.** The counts for the genes detected in *dPLIN2*[-/-] and wild type. The columns from left to right show the Flybase gene name, the estimated counts for *dPlin2* replicate 1, the estimated counts for *dPLIN2* replicate 2, the estimated counts for *dPlin2* replicate 3, the estimated counts for wild type replicate 1, the estimated counts for wild type replicate 2, the estimated counts for wild type replicate 3.
(XLSX)

**S8 Table. GO FlyEnrichr RNAseq.** Gene ontology scores determined by FlyEnrichr for genes which are differentially regulated in all 3 lipid droplet mutant embryos (*Jabba*[-/-], *klar*[-/-] and *dPlin2*[-/-]). The columns from left to right show the gene ontology term, how many of our DE genes match that term out of the total, P-value which indicates if there is a significant relationship between our mutants and this GO term, Adjusted P-value which corrects P-values for multiple testing, Old P-value calculated from a previous version of FlyEnrichr, Old adjusted P-value calculated from a previous version of FlyEnrichr, z-score which is computed using a modification to Fisher's exact test providing rank data, combined score is a combination of the p-value and z-score, and the DE genes matching the GO term.
(PDF)

**S9 Table. Mass spec results WT Jabba and dPLIN2.** The data collected in the mass spec experiment comparing wild type, *Jabba*[-/-] and *dPlin2*[-/-] The columns from left to right show the UniProt accession identifier, the gene name, a protein description, the number of unique peptides matching to gene, median abundance, $Log_2$ Fold change (highlight blue if the change is less than -1 and red if it is greater than 1), P-value (highlight if significant/ <0.05), and the normalized abundance values.
(XLSX)

**S10 Table. Mass spec Gorilla submission lists.** The UniProt identifiers for proteins determined to be hits (differentially abundant *Jabba*[-/-] and *dPlin2*[-/-]) and background (similarly abundant in wild type, *Jabba*[-/-] and *dPlin2*[-/-]).
(XLSX)

**S11 Table. Mass spec Gorilla GO output.** The gene ontology output from Gorilla using proteins differentially abundant in *Jabba*[-/-] and *dPlin2*[-/-]. The columns from left to right show the GO term, the GO term description, the P-value, the false discovery rate/q-value, and the enrichment of the GO term in the data set. Rows in blue show function related GO terms, orange show process related GO terms, and green show location related GO terms.
(XLSX)

**S12 Table. All numerical data.** All the numerical data underlying graphs and summary statistics. Data from individual figure panels are presented on separate tabs. Listed are the numerical data, the means, and the standard deviation.
(XLSX)

**S1 Video. LD motion in wild-type Stage 5 embryos: LDs move bidirectionally along microtubules.** The embryos are alive and have been injected with SPY-tubulin and BODIPY 493/503 to track motion. The videos capture 2 minutes real time. The video field of view shows ~5 blastoderm nuclei coordinating microtubules. The apical region and eggshell are towards the top of the video while the basal region and presumptive yolk cell are towards the bottom. Stills from this movie are in Fig 2C.
(AVI)

**S2 Video. LD motion in *Jabba*[-/-] Stage 5 embryos: those LDs not bound to glycogen show normal motion along microtubules.** Embryo handling, imaging, video parameters, and orientation as for Video S1. Stills from this movie are in Fig 2C.
(AVI)

**S3 Video. LD motion in *klar*[-/-] Stage 5 embryos: LDs move more slowly and for shorter distances than in wild type.** Embryo handling, imaging, video parameters, and orientation as for Video S1. Stills from this movie are in Fig 2C.
(AVI)

**S4 Video. Embryos from *klar*[-/-] mothers struggle with hatching.** Wild type on the left, klar[-/-] on the right. Hatching assay performed on a reciprocal cross for *klar*[-/-] and wild type. The labels on the initial frame correspond to the maternal genotype. The video was captured at 1 frame per minute, and 550 frames are included (i.e., the video is over 9hrs long) at 25˚ C. Wt 1 and 2 are included because they are unremarkable, wt 3 is among the last embryos from the wt mothers to start attempting to hatch (moving), and wt 4 is included because it is among the last embryos from wt mothers to hatch. Klar 1 and 2 are included because they started hatching movements relatively early relative to peers and klar 3–4 started hatching relatively late. Klar 3 hatched after the video ends and klar 1 fails to hatch.
(AVI)

**S1 Graphical Abstract.**
(TIF)

# Acknowledgments

We thank the FlyBase (supported by NHGRI Award #: U41HG000739 and U24HG010859) for annotations and curations. We thank Bloomington Drosophila Stock Center (NIH Award: P40OD018537), the Kyoto Stock Center, Dr. Elizabeth Knust, and Dr. Xun Huang for fly stocks. We thank Dr. J. David Lambert for allowing us to use his mud snail embryos. We thank Dr. Stefanie Schirmeier for her generous gift of antibodies. We thank Dr. Zhihuan Li who initiated inquiry in Jabba's impact on embryogenesis. TEM analysis was performed by Chad Galloway and Karen Bentley in the Electron Microscopy Shared Resource Laboratory at the University of Rochester Medical Center. Mass spec analysis was performed by Kyle Swovick from URMC's Mass Spectrometry Resource Lab. We are grateful for Alicia Shipley's help with Tret1-1 staining. For their insightful input on the manuscript and research strategy, we thank Jonathon Thomalla, Alicia Shipley, Dr. Patrick Sheehan, and Dr. Elizabeth Gavis.

## Author Contributions

**Conceptualization:** Marcus D. Kilwein, Michael A. Welte.

**Formal analysis:** Marcus D. Kilwein, T. Kim Dao.

**Funding acquisition:** Marcus D. Kilwein, Michael A. Welte.

**Investigation:** Marcus D. Kilwein.

**Project administration:** Michael A. Welte.

**Supervision:** Michael A. Welte.

**Writing – original draft:** Marcus D. Kilwein, Michael A. Welte.

**Writing – review & editing:** Marcus D. Kilwein, T. Kim Dao, Michael A. Welte.

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
