## [Decision Letter · Decision Letter 0]

27 Jan 2023

Dear Dr Welte,

Thank you very much for submitting your Research Article entitled 'Drosophila embryos allocate lipid droplets to specific lineages to ensure punctual development and redox homeostasis' to PLOS Genetics.

The manuscript was fully evaluated at the editorial level and by independent peer reviewers. The reviewers appreciated the attention to an important topic but identified some concerns that we ask you address in a revised manuscript.

We therefore ask you to modify the manuscript according to the review recommendations. Your revisions should address the specific points made by each reviewer.

Yours sincerely,

Pablo Wappner

Academic Editor

PLOS Genetics

Gregory P. Copenhaver

Editor-in-Chief

PLOS Genetics

Reviewer's Responses to Questions

**Comments to the Authors:**

Reviewer #1: This study dissects how the asymmetric inheritance of lipid droplets (LD) in the developing Drosophila embryo impacts organismal homeostasis and development. They use specific mutants (Jabba and Klar KO) to affect the inheritance of LDs into cells during embryo cellularization. LD-allocation mutant embryos appear to display defects in LD turnover, development, and delayed hatching. Embryos deficient in perilipin dPLIN2 also display fewer LDs and similar developmental delays. Transcriptomic and proteomic analysis of control and mutant embryos revealed changes in several genes/pathways, including redox homeostasis, sugar metabolism, and mitochondrial metabolism. Of these, proteins involved in glutathione (GSH) redox homeostasis were further investigated. Genetic perturbation of GSTT4 in the Jabba and dPLIN2 mutant backgrounds led to further developmental defects, suggesting an importance of this particular for GST enzyme in the absence of LD homeostasis. Consistent with this, LD-defective mutants display increased lipid peroxidation. LD-allocation mutants also display increased dependence on the ATGL triglyceride lipase.

This is an interesting study that dissects the role of LDs and LD inheritance in Drosophila development. It nicely uses a variety of developed genetic tools to profile how blocking proper LD allocation can impact embryo development. It also implies a protective role for LDs in redox homeostasis in Drosophila. The experiments are generally well conducted. Some additional work dissecting how redox proteins, and in particular ATGL, contribute to organismal development in the absence of proper LD allocation will strengthen the current study.

Specific concerns:

1) In Figure 3A, it is noted that while early stage larva had similar TAG levels, later stage Jabba and Klar KO embryos had elevated TAG levels compared to wildtype. This is interpreted as a defect in LD turnover in these mutants. All these results are compared to sterol levels as a method of normalization. It is generally more common to standardize TAG levels to total protein levels, rather than sterol levels. Are these trends also observed if normalized to protein?

2) The end of the manuscript indicates that GST redox enzymes and the triglyceride lipase dATGL are important for organismal survival in the LD-allocation mutants. The redox enzymes are proposed to be important to protect against lipid/protein oxidation, but the importance of dATGL is less clear. It is proposed that it is needed to consume mis-allocated LDs. However, it seems that dATGL may be needed to supply fatty acids for oxidative metabolism and bioenergetics. Investigating this seems important, since mitochondrial metabolism genes/proteins were also upregulated in the LD-allocation mutants. Does perturbation of mitochondrial fatty acid oxidation also perturb survival in the LD-allocation mutants?

3) Related to point 2, if LD-allocation mutants have increased lipid peroxidation and ROS associated damage, what is the major source of the ROS? It is possibly from alterations in mitochondrial metabolism? Some discussion of this, or some profiling of how mitochondrial metabolism is altered in the LD mutants would mechanistically add to the study.

4) The text is a bit long, and clarity would be improved by shortening the Introduction and Results sections

Reviewer #2: Here Kilwein, Dao, and Welte use the fruit fly Drosophila melanogaster as a model to investigate how the distribution of embryonic lipid droplets alters development and viability. As a first step towards this goal, the authors use mutations in the genes Jabba and klar to determine how disruption of lipid droplet movement alters embryonic development. In these mutant backgrounds, lipid droplets fail to redistribute to peripheral cells and instead move to the yolk cell with glycogen granules. These mutant studies reveal that the mislocalization of lipid droplets results in decreased embryonic lipid consumption as well as developmental delays. A complementary analysis of dPLIN2, Jabba and klar mutants using a combined RNAseq/proteomics approach reveals a link between decreased lipid consumption and glutathione metabolism. Notably, Jabba and PLIN2 mutants are sensitive to loss of GSTT4 and lipid deprived embryos display elevated levels of peroxidated lipids.

Overall, I found the study to be important and potentially interesting. The topic of embryonic lipid distribution is understudied and the authors make novel observations that could serve as the foundation of many future studies. Moreover, the experiments seem well executed, logical, and appropriately interpreted. However, I have a few concerns that should be addressed:

1. Lines 279-284: It would be nice to show how lipid droplet levels change in embryos from the LpR1; Lpr2/+ cross. Either Nile Red or TLC is fine.

2. Lines 443-450: I’m a bit surprised that RNAi depletion of only GSTT4 is sufficient to induce a synthetic phenotype with Jabba and PLIN2. I’m a bit worrited that the RNAi vector targeting GSTT4 would also have off-target effects on other GSTs. The authors should provide evidence that the RNAi TRiP line doesn’t have off target effects. For example, does a TRiP line against a different GST result in a similar sensitivity? I’d note that BDSC 77777 is a viable Trojan insertion in GSTT4, which would allow for a confirmatory genetic experiment.

3. Lines 389-411 – The text completely lacks references. In particular, many of the genes involved in glutathione metabolism have been previously studied in the fly and the text should reference the appropriate literature.

4. Line 412-13 – Again, please cite the appropriate Drosophila literature. For example, Gclc mutations are known to alter glutathione levels (10.1074/jbc.M308035200).

5. In general, data presented in the figures lack an adequate explanation of the statistical test used and what error bars represent. Please include this information in the relevant figure legends. Also, the methods explain that a student’s t-test was used throughout. This test is not appropriate for all of the experiments. For example, Figure 7A is clearly making multiple comparisons and a t-test is not appropriate for this analysis. Also, please check to make sure a parametric test is appropriate for each dataset.

6. Also regarding the figure, some data is present as histograms with no data point, some as histograms with data points, and some as a scatter plot. Please be consistent.

7. In Figure 3C, the data are presented as normalized TAG. The methods state that TAG levels were first normalized to protein using a Bradford assay. If so, the legend and y-axis label should state this.

8. Please acknowledge FlyBase and the appropriate grant number in the acknowledgement section – much of this analysis is made possible by FlyBase. Also, the acknowledgement of the BDSC should include the appropriate grant number.

Reviewer #3: Kilwein et al describe a novel role for redox biochemistry in lipid partitioning and overall fitness of the developing embryo. Lipid droplet transport via Jabba and klar is fundamental and essential for the embryonic developmental rate and fuels the efficient progress of insect (and probably many species) embryogenesis. This seemingly conserved phenomenon is news to me and integrates cytoskeletal mechanics, energy and redox homeostasis, careful developmental staging and microscopy, and omics to elucidate an antioxidant mechanism that compensates for problems in lipid metabolism during development.

This manuscript combines genetics, genomics, proteomics, and biochemistry to produce a major advance in two fields- developmental biology and metabolism. It’s summarized well in the Discussion: “Here we provide the first evidence that LD allocation amongst embryonic lineages impacts subsequent development.”

Major concerns: none. It’s possible to improve the manuscript, but only incrementally.

Minor concerns:

line no.

122: Are these mechanisms completely separate? Consider disparate or another term. Combining the two alleles probably isn’t feasible but could provide information on whether they are independent or whether one is upstream or downstream of the other. Their model suggests that the mechanisms are likely to interact.

472+: The text describes peroxidation of a single (artificial) lipid and this may or may not represent peroxidation of endogenous lipids. C11 is rare in vivo and is exogenously supplied and is unlikely to be in the same subcellular location as the lipids in the droplet. Your genetics experiments certainly support a role for redox balance in these mutants, but it’s more accurate to use “increased levels of BODIPY-C11 lipid peroxidation” as in line 493 instead of implying that the cellular lipids themselves have undergone peroxidation, which may not be true. This would require lipidomics and you haven’t done that.

484: “This oxidative burden does not seem to compromise mitochondrial function” has not been shown. The authors only assayed mitochondrial membrane potential which is not a measure of its function. Mitochondrial function could be measured by the ATP produced or oxygen consumption rate, which would presumably be different if fatty acid oxidation is reduced and glycogen catabolism increased, as they posit. In this case, I would recommend being more conservative with the conclusions drawn.

The Discussion is good and addresses the relevant fields and implications of the work.

The Methods also seem to be appropriate and complete.

Check that genes and mRNAs are in italics. Sometimes -/- is superscript and other times not.

Figures

All figs: should add n to legends. Is n=3 only 3 embryos or 3 groups of embryos. 3 is a small n for Drosophila, at least for experiments that aren’t costly.

Graphical abstract: suggest changing the color of mesoderm to differ from the redox stress color.

One nucleus is outlined in black and the others aren’t. Do you want to label this as a nucleus, or is it even needed for the purpose of this graphic?

What is the second cell type for ectoderm? If this is meant to represent “more than one kind of ectoderm,” that is also true for meso and endoderm origin cells. May want to consider additional labeling or a legend for the graphical abstract.

Fig 1: font is quite small

1A is singular for all species except flies; suggest “fly”

spelling error in 1A “asymetrically” and 1C “micotubules” and “strores”

The descriptions for 1C seem like they should be incorporated into the legend.

How do the stages in 1C correspond to the numbered embryogenesis stages used elsewhere in the manuscript and literature?

Fig 2A: microtubules should connect to the nucleus.

Jabba -/- LDs look larger than controls (and klar mutants) in the micrographs, not smaller as shown in the 2A cartoon.

Fig 3A graph panels should be the same size.

3C: are there significant differences here? Error bars may be missing from some columns.

3D-3E: align these so the Y axes are the same level and make fonts the same size for both

Fig 5A: it looks like a small space between some of the dorsal appendages and vitelline membrane

Fig 5D: colors for down-regulated and the genes don’t match

5E: log base 2 – should make it a subscript

Fig 5F: gene track seems like it belongs at the top next to the gene name. Italicize the gene name.

Fig 6. Differentially expressed is a term that typically refers to RNA levels, including in the previous fig.

Consider the term differentially abundant or differentially present, which better represent the observation made and distinguish the two approaches more clearly for the reader. 6C says “genes which are differentially expressed” and “scaled to the number of genes matching the term” which may be confusing if you used proteins.

Some fonts are very small here. Are all of these graphs needed? The GO and subsequent biochemical analyses make the role for GSH clear.

Fig 7. Please make graph and font sizes consistent, also the colors that denote PLIN2 and Klar are changing throughout the fig, would be better if this were consistent too.

Fig 8. Oregon R isn’t the best paternal control genotype as it’s not matched to the RNAi. These transgenic flies are typically made in a different white-eyed genetic background. It *IS* reassuring that there is no difference between OR and ATGL RNAi in the wild-type maternal background. And there is some pretty elaborate genetics and a lot of data to support their model so that repeating the analyses with another control may not be practical.

8D. Make graphs the same size; genotypes and mRNAs should be italicized in all text and figs

Video 4- the labels for genotypes flash quickly and I missed them the first time.

Consider adding this info to the video legend in lines 1050-1051 “Wild Type on the left, klar -/- on the right”

**Have all data underlying the figures and results presented in the manuscript been provided?**

Reviewer #1: Yes

Reviewer #2: Yes

Reviewer #3: Yes

PLOS authors have the option to publish the peer review history of their article (what does this mean?). If published, this will include your full peer review and any attached files.

Reviewer #1: No

Reviewer #2: No

Reviewer #3: No

---

## [Decision Letter · Decision Letter 1]

17 Jul 2023

Dear Dr Welte,

We are pleased to inform you that your manuscript entitled "Drosophila embryos allocate lipid droplets to specific lineages to ensure punctual development and redox homeostasis" has been editorially accepted for publication in PLOS Genetics. Congratulations!

Yours sincerely,

Pablo Wappner

Academic Editor

PLOS Genetics

Gregory P. Copenhaver

Editor-in-Chief

PLOS Genetics

Comments from the reviewers (if applicable):

Reviewer's Responses to Questions

**Comments to the Authors:**

Reviewer #1: The revised manuscript has addressed the major concerns. Normalization in Figure 3 now accounts for both protein and sterol levels. Concerns with the role of dATGL and mitochondrial fatty acid oxidation are now much more thoroughly discussed, which enhances the study. The study is interesting and timely for the field.

Reviewer #2: The authors have addressed my concerns. I applaud them for characterizing the GSTT4 mutant strain.

Reviewer #3: The authors have done a very good job responding to my concerns and the concerns of the other peer reviewers- and this was already a remarkably solid manuscript at first review. Where they don't change what we've asked for, they seem to have a good reason for it.

**Have all data underlying the figures and results presented in the manuscript been provided?**

Reviewer #1: Yes

Reviewer #2: Yes

Reviewer #3: Yes

PLOS authors have the option to publish the peer review history of their article (what does this mean?). If published, this will include your full peer review and any attached files.

Reviewer #1: No

Reviewer #2: No

Reviewer #3: No

**Data Deposition**

http://datadryad.org/submit?journalID=pgenetics&manu=PGENETICS-D-22-01312R1

**Press Queries**

---

## [Editor Report · Acceptance letter]

4 Aug 2023

PGENETICS-D-22-01312R1 

Drosophila embryos allocate lipid droplets to specific lineages to ensure punctual development and redox homeostasis 

Dear Dr Welte, 

We are pleased to inform you that your manuscript entitled "Drosophila embryos allocate lipid droplets to specific lineages to ensure punctual development and redox homeostasis" has been formally accepted for publication in PLOS Genetics! Your manuscript is now with our production department and you will be notified of the publication date in due course.

With kind regards,

Anita Estes

PLOS Genetics

On behalf of:
